# Modelling the Ruin of Forests under Climate Hazards

Pascal Yiou[1] and Nicolas Viovy[1]

[1]Laboratoire des Sciences du Climat et de l'Environnement, UMR 8212 CEA-CNRS-UVSQ, IPSL & U Paris-Saclay, 91191 Gif-sur-Yvette Cedex, France

**Correspondence:** P. Yiou (pascal.yiou@lsce.ipsl.fr)

**Abstract.** Estimating the risk of collapse of forests due to extreme climate events is one of the challenges of adaptation to climate change. We adapt a concept from ruin theory, which is widespread in econometrics and the insurance industry, to design a growth/ruin model for trees, under climate hazards that can jeopardize their growth. This model is an elaboration of a classical Cramer-Lundberg ruin model that is used in the insurance industry. The model accounts for the interactions between physiological parameters of trees and the occurrence of climate hazards. The physiological parameters describe interannual growth rates and how trees react to hazards. The hazard parameters describe the probability distributions of occurrence and intensity of climate events. We focus on a drought/heatwave hazard. The goal of the paper is to determine the dependence of ruin and average growth probability distributions as a function of physiological and hazard parameters. Using extensive Monte Carlo experiments, we show the existence of a threshold on the frequency of hazards beyond which forest ruin becomes certain in a centennial horizon. We also detect a small effect of strategies to cope with hazards. This paper is a proof-of-concept to quantify collapse (of forests) under climate change.

## 1   Introduction

Extreme events such as droughts and heatwaves are climate hazards that have short and long term effects on forests. Accumulation of drought/heat stress in forests due to recent events might lower their resilience to future extreme events (e.g. Wigneron et al., 2020; Flach et al., 2018; Bastos et al., 2020; Anderegg et al., 2015a). It has been observed that such events increase the chance of tree mortality (Anderegg et al., 2013). This increase of tree mortality probability questions the survival of tree species in some regions of the world (Zeppel et al., 2011; Lindenmayer et al., 2016). The mechanisms leading to tree mortality include complex physiological processes that can depend on the tree species and type of hazards (e.g. Choat et al., 2012).

Most studies on tree death are based on direct or indirect observations of the tree behavior and their growth parameters. They give precious information on the observed global response of forests to those climate hazards. But they are by essence limited to the length of the observation period, and have mostly focused on a few key observed events (e.g. Rao et al., 2019). This can hinder a statistical description that would be necessary to build projections, or estimate *risk* (Field et al., 2012) as a response

to climate change. Here *risk* is understood as the probability distribution of a failure (e.g. irreversible damage or death) due to climate hazards (Katz, 2016).

The potential disappearance of whole forest or a given tree species, due to changes in climate features can be qualified as a *tipping point* of climate change. There is ample literature on tipping points in the climate system, i.e. climate thresholds beyond which ecosystems change behavior (Lenton et al., 2008; Levermann et al., 2012). These papers have defined methodologies to identify climate thresholds beyond which ecosystems are endangered. They have been used to infer tipping points of forests (Reyer et al., 2015; Pereira and Viola, 2018).

The key concept of this paper is the use of the so-called *ruin theory* to provide estimates of the probability of such tipping points. There have been many papers in the econometrics literature that describe ruin models for insurance and finance since the seminal work of Lundberg (1903). A mathematical and statistical framework (Asmussen and Albrecher, 2010) has helped determining the optimal parameters of such models, so that insurers or investors limit the risk of losing their investment and maximize their gain. Ruin models are used to describe the probability that a company that grows with regular income can get bankrupt due to external hazards. To the best of our knowledge, this literature has never been adapted to environmental sciences.

We chose to investigate trees or forests because trees are adapted to live for a long time and can survive climate adversity that can occur during a given year, contrary to annual plants. Process-based growth models for trees can be devised (e.g. Han and Singh, 2020) and observations of tree mortality are available (e.g. Choat et al., 2012). Tree growth is affected by climate variations in various ways. Heatwaves and droughts can alter (and lower) tree reserves and the capability for growth during the years following an extreme climate event, which affects their average growth and can increase their chance of mortality (von Buttlar et al., 2017; Sippel et al., 2018). For this purpose, trees contain a large amount of non structural carbohydrates (NSC). These NSC reserves allow to rapidly build a large surface of leaves (potential productivity is related to this foliage area) at the time of leave onset during the next year. NSCs also help renewing the leaves in case of defoliation (related to frost or insects, for instance) and, more generally, provide the necessary energy for metabolism, including protection from pathogen diseases. Hence a part of annual productivity is devoted to accumulate carbohydrate reserves (i.e., larger than what is necessary for the next year initiation of the leaf phenological cycle). As there is a competition in allocation of assimilates between NSC and plant growth, the plant does not accumulate reserves indefinitely, but tends to reach an optimum value (Barbaroux et al., 2003). Climate hazards can reduce the quantity of assimilate allocated to the reserve (He et al., 2020). This can be caused directly by reducing productivity. But climate hazards can also indirectly cause plant damage like xylem embolism. This embolism can kill trees in extreme cases (especially for young trees). This process kills only parts of mature trees, which affects the total amount of NSC, and also affects the development a new foliage and jeopardizes the ability to grow during the following years. So even though this effect does not directly impact NCS reserves, it induces equivalent consequences for tree growth in years following climate hazards. Several studies have investigated the physiological processes leading to tree mortality based on empirical observations (Adams et al., 2009; Bigler et al., 2007; Bréda and Badeau, 2008; Villalba and Veblen, 1998; Matusick et al., 2018). These studies compare relative tree ring growth between dead and healthy surrounding trees with same diameter during the years prior to tree death. This comparison, by removing annual climate effects, is then a good proxy of the difference

in NSC reserves. These studies conclude that dying trees have a systematically lower level of reserves than for living trees. Hence, even if trees are not fully depleted of NSC, there is a critical level of NSC under which trees cannot recover and then die. These studies suggest that the levels of NSC in trees are good indicators of possible forest die-back. Allocation to NSC reserves can hence be compared to an insurance system when each year, a part of the productivity is "paid" to the insurance (i.e. NSC for trees) but in return can be mobilized in the case of damage. Then "ruin" occurs when the level of NSC reaches a critical level under which the probability of tree death on the short term becomes very likely. This loosely justifies a parallel between insurance and tree systems.

At present, full size process-based models of tree growth forced by large ensembles of climate simulations (e.g. Massey et al., 2015) require large computing resources that limit the number of numerical experiments that can be performed to reliably estimate probability distributions of key environmental variables. This paper addresses this limitation and uses a framework based on the Cramer-Lundberg toy model to generate probability distributions of climate hazards.

The goal of this paper is to formulate a simplified model of tree growth and the impacts of climate hazards, which can be interpreted as a ruin model, and use its multiple simulations in order to evaluate its sensitivity to hazard parameters. Our study investigates the probability of "ruin" of a population of trees that are subjected to heat and drought stress. In this context, tree ruin is reached when its carbon reserves fall below a critical threshold, meaning that the growth of trees is no longer guaranteed. In principle, this concept of ruin could be applied to a forest in general, but also to a given tree species, which means that this species will disappear from the forest but that a forest with other more tolerant species can still survive. Such an application requires the determination of key physiological parameters that are adapted to the species, and the determination of climate hazard statistical parameters.

Section 2 introduces a growth/ruin model for trees based on the Cramer-Lundberg ruin model, and discusses the interpretation of its parameters (section 2.1). Section 2.2 introduces a simple tree growth model based on the Cramer-Lundberg ruin model for insurances (Embrechts et al., 1997). We investigate its properties in order to evaluate the occurrence of ruin (i.e. when trees stop growing) within a fixed time horizon, due to the impacts of extreme events, under various climate scenarios. Here, the term "horizon" is the maximum (finite) period for which simulations are made, which in practise, can be viewed as the end of the 21st century. This approach is meant to tackle quantitatively the issues of tipping points of ruin for a specific field (forestry), although this concept could be extended to other domains. Here, *quantitative*, implies that the probability distributions of key ruin parameters (e.g., time of ruin and average carbon reserve) are determined explicitly. Unlike the Cramer-Lundberg ruin model, our tree model cannot be solved analytically and extensive numerical simulations are therefore used. A standard sample for the insurance industry has to contain more than $10^4$ members for robust probability estimates. In section 3 we detail the meteorological data that are used to construct a damage function (due to climate hazards). Section 4 explains the experimental protocol for the analyses. The results and interpretations are developed in section 5. An Appendix is devoted to the development of a drought index based on precipitation and temperature.

## 2 Methods

### 2.1 Cramer-Lundberg ruin model

This section introduces the key concepts of ruin models. The insurance industry uses such statistical models to determine the premium rates in order to avoid ruin (i.e., bankruptcy) when hazards occur. Those statistical models are based on the simple Cramer-Lundberg model (Asmussen and Albrecher, 2010; Embrechts et al., 1997), which can be formulated as:

$$R(t) = R_0 + p \cdot t - S(t), \tag{1}$$

where $R(t)$ is the insurance capital at time $t$, $R_0 > 0$ is the initial capital, $p > 0$ is the premium rate that is collected every year $t$, and $S(t) \geq 0$ is a damage function that represents the (random) losses to hazards that occur up to time $t$. The only random part of the model in Eq. (1) stems from the damage function $S(t)$. A ruin is declared and the process is stopped when $R(t) \leq 0$. If $S(t)$ is always 0, then the capital grows indefinitely. The literature on ruin theory describes how the probability distribution of $R(t)$ depends on hypotheses on hazards conveyed by the random damage function $S(t)$.

One can be interested in the behavior of the system before a finite horizon $T > 0$, e.g. a few decades. We define the ruin probability $\Psi$ before horizon $T$ by:

$$\Psi(R_0, p, T) = \Pr\left(R(t) \leq 0, \text{ for some } T \geq t > 0\right) \tag{2}$$

and the ruin time $\tau(R_0, p, T)$ by the first positive time when ruin is reached:

$$\tau(R_0, p, T) = \inf\{t > 0, R(t) \leq 0\}, \tag{3}$$

where $\inf A$ is the infimum value of the set $A$. Since $S(t)$ is a random variable, we are interested in $E(\tau)$, the expected value of $\tau$ with respect to the random variable $S(t)$. If ruin never occurs during simulations of $R$, then $E(\tau) = \infty$. Actors in insurance companies try to estimate the smallest premium rate $p$ from expert knowledge of the probability distribution of $S(t)$ in order to avoid ruin, as it is assumed that the lowest $p$ would make them more attractive to clients. This gives them a lead in the competition against more greedy companies (which are subject to the same damages $S(t)$).

In several instances, there is no acceptable value for $p$ to prevent from ruin, i.e. the expected value of $\tau$ can be lower than the value $T$ (say, with some low probability). This is why insurance companies resort to re-insurance to avoid bankruptcy after unexpectedly large losses $S(t)$. Re-insurance companies are insurers of insurance companies (e.g. https://en.wikipedia.org/wiki/Reinsurance). There is no obvious re-insurance mechanism for natural systems.

There is no re-insurance mechanism for trees, other than human intervention.

The damages $S(t)$ are generally represented as a *random sum* of random variables:

$$S(t) = \sum_{k=1}^{N(t)} X_k, \tag{4}$$

where $N(t)$ is a Poisson random variable that accounts for the number of hazards occurring up to time $t$, and $X_k$ are random variables that account for the cost of each hazard. A Poisson distribution is used to express the number of events that occur

over a fixed period of time. Hence, the probability distribution of $N(t)$ can be written:

$$\Pr(N(t) = n) = \frac{\lambda^n}{n!} e^{-\lambda}, \tag{5}$$

where $\lambda > 0$ is called the intensity of the Poisson distribution. The mean and variance of the Poisson distribution are $\lambda$.

The probability distribution of $X_k$ can be modeled by an extreme value law, like a generalized Pareto distribution (GPD) (Coles, 2001; Embrechts et al., 1997), when a hazard variable exceeds a high threshold $u$. The GPD describes the probability distribution of a random variable $X$ when its value exceeds a threshold $u$. Its cumulative probability distribution function is:

$$\Pr\{X > x | X > u\} = \left[ 1 + \xi \left( \frac{x - u}{\sigma} \right) \right]^{-1/\xi}, \tag{6}$$

where $\sigma > 0$ is a scale parameter and $\xi$ is a shape parameter that states how fast extremes grow. The parameters of the Poisson distribution for $N(t)$ and the GPD distribution are estimated from prior information, e.g. observations or expert knowledge.

There is ample statistical literature in finance on the relation between $\tau$ and the probability distribution of $S$ (Embrechts et al., 1997; Asmussen and Albrecher, 2010). In practice, estimates of $E(\tau)$ or an optimal $p$ can be obtained by simulating the model of Eq. (1) and estimating empirical probability distributions.

The notion of finite horizon $T$ is useful when considering that an investment (in the insurance sector) is made for a finite time. We will be interested in generating many finite sequences of $S(t)$, corresponding to a sample of all possible $T$-long trajectories.

## 2.2 A ruin model for trees

The goal of this sub-section is to adapt the Cramer-Lundberg model in Eq. (1) to formulate a simple tree growth model that explicitly takes into account a damage $S(t)$ due to a climate hazard, such as drought and heatwave.

Here $R(t)$ is the amount of non-structural carbohydrates in a tree (hereafter called reserves, in kg C/m$^2$) that allow tree growth at the beginning of the growing season, e.g., the start of spring in the northern hemisphere. We assume that trees spend a fraction of their reserves to grow their different plant organs (e.g. roots, stem, leaves), depending on their previous state. As stated previously, we also assume that tree resources are bounded by a maximal value $R_{\max}$ (Barbaroux et al., 2003). The value of $R_{\max}$ sets the scaling of the reserve $R(t)$. The value of $R_{\max}$ is estimated to be around 2 kg C/m$^2$ for beech and oak trees (Barbaroux et al., 2003). In order to keep genericity (the model of ruin does not depend on the exact value of $R_{\max}$) and simplicity, we normalize the reserve value $R(t)$ by $R_{\max}$ so that, the reserve will be expressed in percentage of $R_{\max}$, with values between 0 and 100 %.

There have been observations of legacy effects of drought hazard on the tree growth during the year after the drought (Anderegg et al., 2015b). This effect depends on the tree species. Because of this decrease in net primary production (NPP) due to drought, we assume that it also affects the allocation to carbohydrate reserves. Hence, the yearly NPP allocated to reserves $p(t)$ (in kg C/m$^2$/year) depends on the climate hazard damage that occurred during the previous year $S(t-1)$ (in kg C/m$^2$):

$$p(t) = p_0 - BS(t-1)/\Delta t, \tag{7}$$

where $p_0$ is the optimum average yearly NPP of a population of trees allocated to NSC reserve, $B \geq 0$ is a memory factor of the damage function, and $\Delta t = 1$ year. More generally, $p_0$ represents the fraction of NPP allocated to $R$ when NPP is itself optimum. But it has been observed from in-situ measurements that the total NPP decreases the year after a hazard (Anderegg et al., 2015b). In general, this is related to the fact that plants have lower leaf area and/or and increased respiration cost related the investment to repairing tissues or defensive costs. So Eq. (7) can be interpreted by the fact that the total amount of carbohydrate allocated to reserves decreases because of the decrease of total NPP ($B$ and $S$ are positive), assuming that the fraction allocated to $R$ is unchanged on average. In principle, the value of $p_0$ can depend on the tree species or location.

We hence introduce a new growth/ruin model for trees that describes the carbon reserve $R$ variations with time:

$$R(t) = \min\left[(1-b)R(t-1) + p(t)\Delta t - S(t), R_{\max}\right], \tag{8}$$

where $b \geq 0$ is the fraction of previous resources (at time $t-1$) devoted to growth, and $\Delta t$ is the time interval (here $\Delta t = 1$ year). In this model, the parameters $b$, $B$ and $R_{\max}$ are called *physiological* as they describe tree growth. We define *tree ruin* when $R = 0$, i.e. when NSC value reaches a critical low level where tree death become almost certain. Taking a positive threshold value (e.g. a small percentage of $R_{max}$) would not change qualitatively the results. We recall that all physiological parameters were normalized to $R_{\max}$, and are therefore expressed in terms of fraction ($b$) or percentage ($p(t)$ in %/year) of the actual $R_{\max}$. The formulation of Eq. (8) is based on an *iterative* version of the Cramer-Lundberg model in Eq. (1).

In this paper, we assume that the type of hazards that can affect tree growth (or survival) are summer droughts (Allen et al., 2010; Choat et al., 2012; DeSoto et al., 2020). In Europe, major summer heatwaves are often concomitant with droughts. This combination of climate factors creates stress for trees, which lowers their NPP and can destroy branches and leaves and impacts their growth and reserve. Other types of hazards could also be considered (storms, pests, etc.).

The hazards do not necessarily occur every year: they arrive at years $t$ that can be modeled by an exponential distribution with parameter $\Lambda$ (Coles, 2001). Thus we assume that the inter-arrival times follow a Poisson distribution with a mean value of $\theta = 1/\Lambda$ (the average return time of hazards). This description is rather generic has been widely used in atmospheric sciences or statistical climatology (e.g. von Storch and Zwiers, 2001; Smith and Shively, 1995). This implies that the number of hazards up to a year $t$ $N(t)$ follows a Poisson distribution with a parameter $\Lambda$, so that this formulation is consistent with the Cramer-Lundberg model (Eq. (1)). When climate hazards occur (at random times), the corresponding damage $S(t)$ during year $t$ is written as:

$$S(t) = A_h \sum_{k=1}^{M(t)} Y_k. \tag{9}$$

Eq. (9) is slightly different from Eq. (4) in which hazard equates to damage, and the damage in Eq. (4) is the cumulated damage up to year $t$. This subtle difference stems from the iterative nature of Eq. (8), which generalizes the direct formulation of Eq. (1). In Eq. (9), the $Y_k$ are the hazards that occur during the year $t$. Hence, the sum of $Y_k$'s in Eq. (9) corresponds to a damage $X$ in Eq. (4). $A_h$ is a normalizing constant that translates the climate hazard conveyed by $Y_k$ into damage $S(t)$. $M(t)$ is the number of hazards (e.g. the number of very hot/dry days) during year $t$ and follows a Poisson distribution with parameter $\lambda$. $Y_k$ are inferred from climate variables like a drought index for heatwaves or wind speed for storms during hazards. We assume

that they follow generalized Pareto distributions (GPD), with scale parameter $\sigma$ and shape parameter $\xi$. The GPD describes the probability distribution of $X_k$ when it exceeds a high threshold $u$ (Coles, 2001). The parameters of the GPD distribution ($\sigma$ and $\xi$) and parameters of the Poisson distributions ($\lambda$ and $\Lambda$) are called the *hazard* parameters.

When no hazard occurs, $S(t) = 0$. If $b = 0$ (no use of reserves for growth), $B = 0$ (no memory of previous hazard) and $M(t) = 1$ (only one hazard at a time at most), then the model in Eq. (8) simplifies to the Cramer-Lundberg ruin model (Eq. (1)), in which $N(t)$ follows a Poisson distribution of parameter $\Lambda$. If hazards never occur (i.e. $S = 0$ at all times), then $R(t)$ converges to $(1 - b)R_{\max}$.

The parameters of the Poisson distributions for $M(t)$ and the Pareto distributions for $Y_k$ can be estimated experimentally from meteorological observations or climate model simulations. The growth parameters $p_0$, $b$ and $R_{\max}$ in Eq. (8) can be obtained from tree physiology databases (Allen et al., 2010; Cailleret et al., 2017) and should be adapted to tree species.

The difficult part is to estimate scales for the values of $A_h$ and $B$. It has been observed that tree species can have differing strategies to face heat and drought stress (Adams et al., 2009; Teuling et al., 2010). The first strategy is used when some tree species grow in spite of the hazard during year $t$. This can be achieved by maintaining stomatal aperture to maintain photosynthesis (anisohydric mechanism) increasing the risk of embolism (Mitchell et al., 2013) or changing allocation to maintain growth of branches and root at expense of carbohydrate reserves (van der Molen et al., 2011). In both cases, these trees "pay" the next year, even if there is no hazard because plant growth occurs at expense of foliage surface and plant protection for next year (van der Molen et al., 2011). This strategy implies that trees have an interannual memory of hazards. Conversely, a second strategy is used by other tree species that stop growing during hazards (by stomatal closure to avoid embolism (isohydric mechanism) or maintain allocation to reserve at the expense of other pools). For these trees, the growth is impacted during the year of the hazard, but this hazard has fewer impacts during the next year. This strategy implies that trees do not have interannual memory of hazards. Those two strategies are more general than anisohydric vs. isohydric mechanisms, as they also include possible change in allocation to growth/reserve. In order to use a botanical terminology and for simplicity, we will use the terms anisohydric/isohydric as they are widely used to describe different responses to drought. It is possible to represent the different strategies in the model through the parameter $B$. The anisohydric strategy, which maintains growth during the year of hazard with impacts on later years (i.e. with interannual memory), can be represented by positive value of $B$ in Eq. (7). Conversely the isohydric strategy, in which growth is reduced during the year of the hazard with less impacts in later years (i.e. no interannual memory), can be represented by $B$ close to 0 in Eq. (7). We will investigate the sensitivity of the ruin probabilities to these tree strategies. To simplify the terminology, we make a parallel with the insurance system: if $B = 0$, trees pay "cash" on their reserve ($S(t)$ then is large when hazard occurs); if $B > 0$ trees allow for a "credit" on the next year ($S(t)$ is reduced in the current year but this strategy implies the possibility of using the reserves the next year). Mitchell et al. (2013) notice that there is a continuum between the two strategies which can be represented by different values of $B$. The values of $B$ are chosen so that the average value of damages (i.e., the expected value of $(1 + B)S(t)$) is a constant. This constant gives the scale of the impact parameter $A_h$.

In this paper, the values of $A_h$ and $B$ are arbitrarily chosen to scale with expected behavior of trees. The range of those parameters could be estimated from in situ observations or expert knowledge. In this proof-of-concept paper, these two parameters are considered to be normalized and do not have units.

## 2.3 Sample trajectories

For simplicity, we normalize all the parameters to the optimum level of reserve $R(t)$ fixed the an arbitrary value of 100. Therefore, $R_{\max} = 100$. $R(t) = 0$ is the value for tree ruin. As explained in introduction, it does not necessarily mean a totally depleted NSC reserve, but a level for which the probability of tree recovery becomes very unlikely even for good weather condition, and trees will die in the short term. (Barbaroux et al., 2003) evaluated the seasonal evolution of NSC for beech and oak trees. The amount of NSC in July (which correspond to the minimum of the NSC cycle) reaches 75% of its maximum. Hence we assume that annual allocation and use of the NSC is 25% of the total (i.e. $p_0 = 25$ %/year and $b = 0.25$). Likewise He et al. (2020) evaluated the impact of several level of droughts on total NSC. They estimate a decrease of NSC of 20% in case of large drought. Therefore, we assume $A_h = 1.2$ so that the average value of damages is $\approx 20\%$ of $R_{max}$. The hazard parameters are $\sigma = 0.1$, $\xi = -0.2$ and threshold $u = 1$ (from the GPD distribution of a drought index), and $\lambda = 10$ days and $\Lambda = 5$ years for the hazard arrivals. The model was run with memory parameter values of $B = 0$ (isohydric or "cash") and $B = 0.4$ (anisohydric or "credit"). We simulated $10^4$ trajectories with those parameters. For each ensemble, we computed the average of the reserve function $R(t)$ before it reaches $R(t) = 0$. We selected four trajectories with the 95th, median and 5th quantiles of the average reserve function, and one of the trajectories with a ruin (i.e. ruin time is $\tau < 100$ years).

Figure 1a and 1b shows the time series of the damage $S(t)$ and the reserve $R(t)$ functions for those four key trajectories, when $B = 0$ (isohydric). As the shape parameter $\xi$ is negative, the damage values $S(t)$ do not yield a large variability (i.e., unbounded). The statistical properties of all trajectories of $S(t)$ are supposed to be similar, as they are drawn from the same underlying distributions. Therefore, we chose to show only two trajectories for $S(t)$: one that achieves ruin (in red) and the one with the median value of the $R(t)$ mean. In this experiment, the reserve averages for the 5th, median and 95th quantile trajectories are respectively 74, 84 and 88 reserve units.

Figure 1c,d shows the time series of the damage $S(t)$ and the reserve $R(t)$ functions for four key trajectories, when $B = 0.4$ (anisohydric). The damage function yields the same statistical properties but the scaling is different so that the integrated damages are similar in both cases: the hazard is scaled so that the damage at year $t$ is distributed over $t$ and $t+1$. For this sample of simulations, the reserve averages for the 5th, median and 95th quantiles trajectories are respectively 75, 84 and 88 reserve units. These values for the anisohydric strategy are similar to those for the isohydric strategy in Figure 1a,b. The reserve $R(t)$ with $B = 0.4$ yield a slightly slower variability for the upper quantiles (black and blue lines in Fig. 1b,d), which is explained by the memory induced by $B > 0$. The empirical probability of reaching a ruin within 100 years (see Eq. (2)) is higher for $B = 0$ ($\Psi = 4 \cdot 10^{-3}$) than for $B = 0.4$ ($\Psi = 1.5 \cdot 10^{-3}$). In both cases, ruin is a rare event for the chosen physiological parameters, and the return period of a ruin (i.e. the inverse of its probability) is larger than 100 years. This justifies our statistical approach of simulating a very large ensemble of trajectories, as the key difference between tree strategies lies on small probability that is difficult to assess from observations.

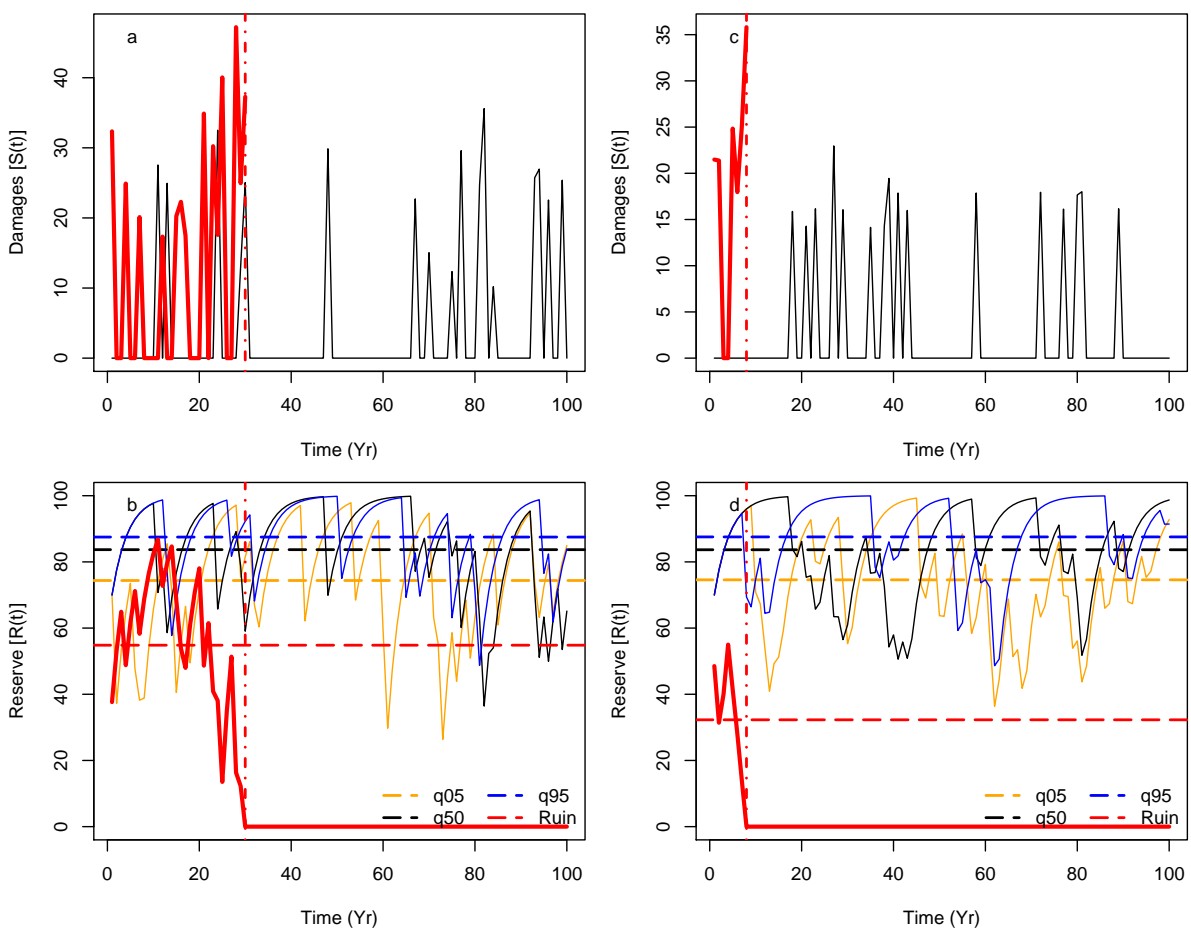

**Figure 1.** Sample of time series of simulations of $S(t)$ (upper panels) and $R(t)$ (lower panels) for $B = 0$ (isohydric: panels a and b) and $B = 0.4$ (anisohydric: panels c and d). The red lines show sample trajectories of $S(t)$ and $R(t)$ for which $R(t)$ reaches 0 for $t < 100$ years. A vertical dashed-dotted line indicates the time of ruin. The black lines indicate trajectories of $S(t)$ and $R(t)$ (for $B = 0$ and $B = 0.4$) that achieve the median of the temporal mean of all $10^4$ trajectories of $R(t)$. The orange and blue lines in panels b and c show the trajectories of $R(t)$ that achieve the 5th and 95th quantiles of all $10^4$ simulation means. The horizontal dashed lines of the lower panels (b and c) represent the means of the represented trajectories, for times before ruin.

A physical evaluation of the model is in principle difficult. The main reason stems from the nature of the phenomenon that is modeled, which does not occur very often. The ruin of insurance companies hardly ever occurs for a complete validation of the Cramer-Lundberg model, and this could be a heuristic argument that it works (at least for that domain of application). In practice, a lot of data have been collected from dead trees and from surrounding trees still alive to evaluate how dead trees behave in the years before they die (e.g. Villalba et al. (1998), Bréda and Badeau (2008)). Hence differences of tree rings width between dead and living trees can be considered as a good proxy of tree reserves. Cailleret et al. (2017) synthesized data from

255

differential tree growth and used it as a proxy for NSC, under various climate hazards. There is a good agreement between our simulated evolution of reserves in case of ruin (Figure 1b,d) and observed evolution of relative tree ring growth before tree death. In particular Cailleret et al. (2017) show that in a majority of cases, there is an observed relative decrease of growth between twenty to fifty years before death, which is consistent with our simulations.

## 3 Data

The goal of this section is to provide climate constraints on the parameters of the hazard function $S(t)$.

### 3.1 Observations

Meteorological data were taken from the European Climate Assessment and Dataset (ECA&D) database (Haylock et al., 2008). We used daily maximum temperature (TX) and daily precipitation (RR) from Berlin, De Bilt, Orly, Toulouse and Madrid stations. This choice was motivated to cover a large area over western Europe. We considered data from 1948 to 2019 ($> 70$ years of daily data). These data had less than 10% missing values.

### 3.2 Drought/heatwave damage index

We consider the drought/heatwave index $I_{YV}$ defined in the Appendix A, based on precipitation frequency and temperature from the ECA&D database. This index is computed over five ECA&D stations (Berlin, De Bilt, Orly, Toulouse and Madrid). The necessity to consider this new index emerged from shortcomings of already existing indices that prevent from a reliable and relevant statistical modeling. On the one hand, the most physically relevant indices are based on soil moisture, but they do not cover a period that it long enough to determine the GPD parameters of its extremes. On the other hand, indices that use well measured meteorological variables (e.g. temperature and precipitation) are not fully adapted to reflect drought (see Appendix A). Therefore, this justifies the development of an index from which we can infer the GPD parameters of the damage function due to hazards. The $I_{YV}$ index yields high values (more than 1.1°C) during summer drought/heatwave events, and low values for wet/cold summers (less than 1°C).

From the Spring-Summer variations of this index, we determine the Generalized Pareto Distribution parameters of the daily $I_{YV}$ index, when the (daily) values exceed the 95th quantile. As the daily values are temporally correlated (by construction, as the index is a running sum), we consider the maxima of clusters above the 90th quantile and determine the number of days that exceed the 95th quantile threshold. This procedure is advocated in the textbook of Coles (2001).

The values and standard errors of the GPD parameters (threshold, scale and shape), and the Poisson parameters for the duration and return periods of events are shown in Table 1. The values of the GPD parameters are consistent with the existing literature (e.g. Parey, 2008; Kharin et al., 2013).

We considered the range of parameters obtained from the 5 stations and their uncertainties, and took the envelope of those parameters from their lower and upper uncertainties. This provides a conservative estimate of the range of variation for the

**Table 1.** Range of values of parameters for the damage function $S(t)$ from the $I_{YV}$ drought/heatwave (HW) index computed for the five stations. The GPD thresholds are the averages of the 95th quantiles of the index for the five stations. The GPD scale and shape parameters are the averages of the estimates on the indices for each of the five stations. The ranges of the GPD parameters take the uncertainties into account.

| Parameter | Interpretation | Average value | range |
|-----------|----------------|---------------|-------|
| $u$ | GPD threshold | 1.7 | $[1; 5]$ |
| $\sigma$ | GPD Scale | 0.15 | $[0.08; 0.2]$ |
| $\xi$ | GPD shape | $-0.32$ | $[-0.45; -0.2]$ |
| $\lambda$ | Nb. dry days | 15 | $[2; 30]$ |
| $\Lambda$ | Return period of HW events (years) | 8 | $[2; 15]$ |

parameters (i.e. a wide range). From those ranges of parameters, we can simulate Generalized Pareto distributions for the damage function in the model of Eq. (9).

## 4   Experimental design

The physiological parameter in Eq. (8) are fixed to $b = 0.25$, $p_0 = 25$ %/year and $R_{\max} = 100$ (in % of $\approx 2$ kg C/m$^2$). We simulate $N = 10^6$ trajectories of $R(t)$ of 100 years, with an initial condition of $R(0) = 60$. For each trajectory, the parameters of the damage function $S(t)$ are randomly sampled with a uniform distribution with a range that is estimated from the heat/drought stress index $I_{YV}$ (in Sec. 3). The bounds of the uniform distributions are given in Table 1 and reflect the variability across Berlin, De Bilt, Orly, Toulouse and Madrid.

From those ensembles of simulations trajectories we determine the average of reserve $R(t)$ before ruin $\langle R \rangle$, and the time of ruin $T_{\text{ruin}}$ (if it ever occurs). By construction of the model, $R(t)$ evolves between 0 and 100 (the optimal reserve) and $T_{\text{ruin}}$ is between 1 (immediate ruin) and 100 (no ruin within 100 year simulations).

The large number of trajectories ($10^6$) helps investigate the dependence of $\langle R \rangle$ and $T_{\text{ruin}}$ on the parameters of $S$, namely $\sigma$, $\xi$, $\lambda$ and $\Lambda$ (see table 1).

## 5   Results

The dependence of the ruin time on the four damage parameters is shown in Figure 2 when $B = 0$. Each boxplot depicts the probability distribution of ruin times for a given value of a parameter, and a random combination of other parameters.

Figure 2 highlights the fact that the system can shift from a "no ruin" state to a probable ruin in a century, with rather small parameter changes of the frequency of extreme days in the $I_{YV}$ index (either the frequency of dry/hot summers, or the number of dry/hot days during a hot summer). The dependence on the scale parameter $\sigma$ and the shape parameter $\xi$ is rather weak (Figure 2, panels c and d). The prescribed range of variations of those GPD parameters is small in absolute values. The GPD

threshold $u$ has an important impact of the ruin time, as the probability distribution of ruin times shift from a median on 100 years to a median value of 70 years within a 6% change of the threshold $u$ (from values of $u = 2.5$ to $u = 2.75$; Figure 2e).

We define a bifurcation of ruin probability when the median ruin year $\tau$ becomes less than 100 years. From this experiment ($B = 0$), we find that a "no ruin/ruin" bifurcation occurs when a return period threshold of 9 years is crossed (Figure 2a) is crossed. If we focus on the last 20 years in western Europe, extreme summer heatwaves and droughts occurred in 2003, 2006, 2018 and 2019. This might imply that European forests with trees that yield those physiological parameters are close a threshold of positive ruin probability.

The threshold on the number of hot days per summer is 14 days (Figure 2b). This parameter controls the magnitude of the random sum in $S(t)$, because the daily hazards $Y_k$ yield a bounded tail ($\xi < 0$). This means that if the length of heatwaves can exceed 14 days, tree ruin becomes significantly likely before the end of the 100 years. Such an event occurred in 2018 in Europe (Yiou et al., 2020).

The probability distribution of the average reserve before ruin $\langle R \rangle$ is shown in Figure 3 for trees with $B = 0$. This figure
highlights that $\langle R \rangle$ weakly depends on the GPD parameters of the damage function (Figure 3cd). The average reserve strongly depends on the return period of heatwaves and the number of hot days during heatwaves and the GPD threshold $u$ (Figure 3abe).

Figure 3a shows that when damages (due to droughts/heatwaves) occur too often (low return periods of events), then the trees do not have enough time to build enough reserves to face the next extremes.

The behavior of tree growth when $B = 0.4$ (anisohydric) is qualitatively similar to the one with $B = 0$ (isohydric) (Figure 4). The bifurcation thresholds for $\Lambda$ is 6 years, 16 days for $\lambda$, and a GPD threshold $u$ value of 3. This suggests a higher resilience of trees with interannual memory ($B > 0$), as the bifurcations occur for higher values of $\Lambda$, $\lambda$ and $u$.

In a second set of experiments, we maintain the scale and shape parameters constant: $\sigma = 0.1$ and $\xi = -0.3$. The other hazard parameters are randomly sampled within the range as indicated in Table 1. As observed in Figure 2ab, a transition from
a median ruin time of 100 years (i.e. no or unlikely ruin) to a ruin time $\tau < 100$ years appears for return times of events $\Lambda$ between 6 and 10 years, for duration of events $\lambda$ between 12 and 16 days, and for a GPD threshold $u$ between 2.25 and 3. Therefore, we focus on the probability distributions of ruin times and reserve near those thresholds, for the isohydric ($B = 0$) and anisohydric ($B = 0.4$) simulations.

Figure 5 summarizes the probability distributions of ruin times and reserve for simulations for all simulations, and simu-
lations with hazard values $\Lambda \in [6, 10]$ years, $\lambda \in [12, 16]$ days, and $u \in [2.25, 3]$. On the whole, the probability of ruin $\Psi$ for $B = 0$ is $\approx 0.53$, and it is $\Psi \approx 0.47$ for $B = 0.4$. Figure 5a shows that the anisohydric simulations ($B = 0.4$) yield a larger median value of ruin time (100 years vs 74 years for $B = 0$). This means that an anisohydric strategy leads to a longer life expectancy. The differences in the reserve are shown in Figure 5b. A Kolmogorov-Smirnov test (von Storch and Zwiers, 2001) indicates a significantly higher value of average reserve for $B = 0$ (83 vs. 81, $p$-value $< 10^{-15}$). The differences amount to $\approx 2$
units of reserve, which is small compared to the optimal value $R_{\max} = 100$.

On the whole, those results show the dependence of the ruin time on the coping strategy: smoothing the damages over two years increases the time to ruin, for various scenarios of extremes (return time, length and intensity). However, the response of

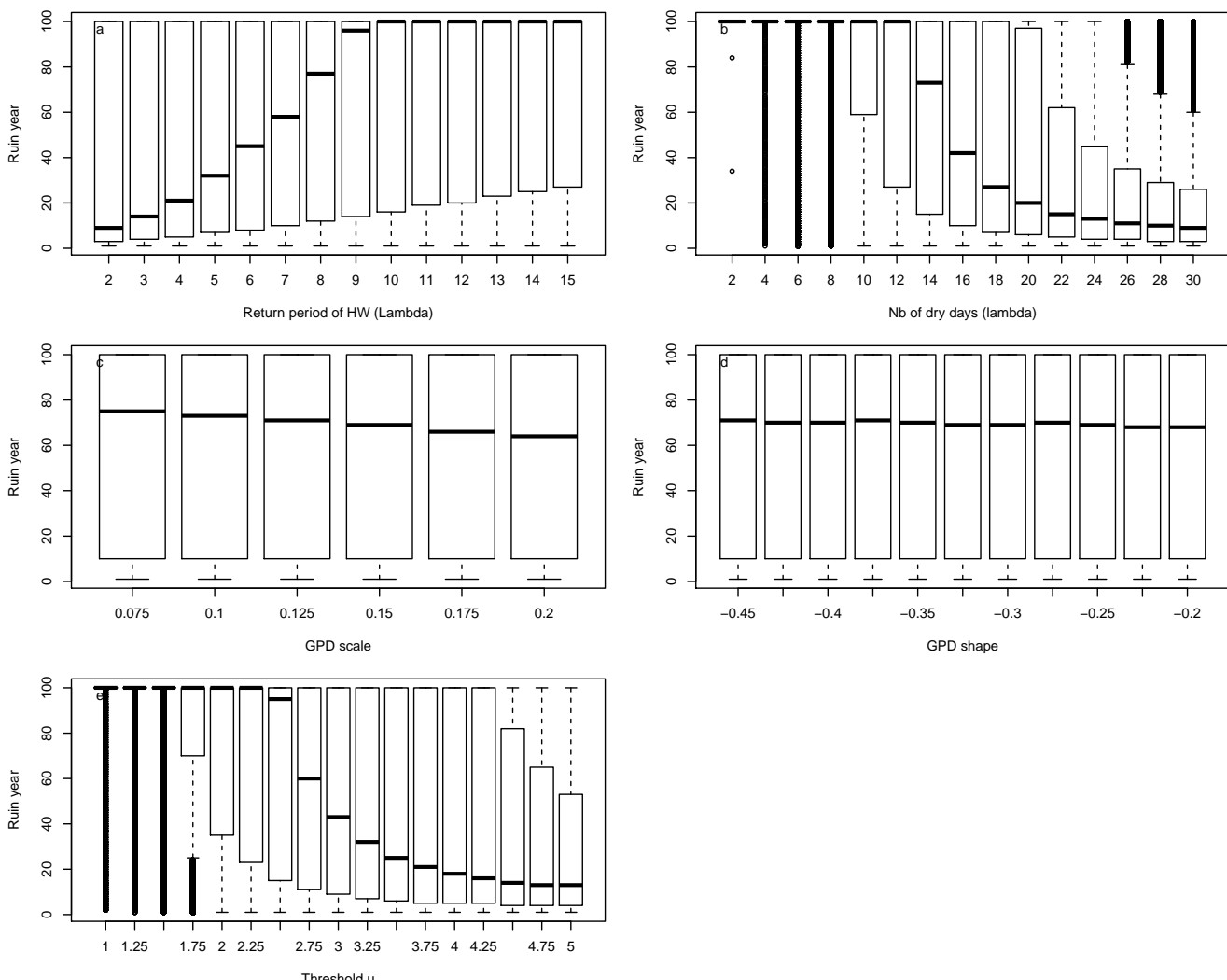

**Figure 2.** Dependence of probability distributions of ruin year $\tau$ as a function of drought/heatwave (HW) return periods $\Lambda$ (a), number of extreme days of $I_{YV}$ days during summers $\lambda$ (b), GPD scale $\sigma$ (c) and GPD shape $\xi$ (d), and threshold $u$ above which damage $S(t)$ is triggered (e). Experiments for $B = 0$. For each value of the control variable, a boxplot is given. The horizontal thick bar of boxplots represents the median ($q50$) of the distribution. The boxes boundaries represent the 25th quantile ($q25$) and the 75th quantile ($q75$). The upper whiskers are $\min[\max(\tau), 1.5 \times (q75 - q25) + q50]$. The lower whisker has the symmetric formulation. The points are for data that are above or below the whiskers.

the reserve depends on the parameters of the generalized Pareto and Poisson distributions: a higher frequency (or lower return period $\Lambda$) favors isohydric strategies, while the intensity (linked to the duration or highest value) slightly favors anisohydric strategies.

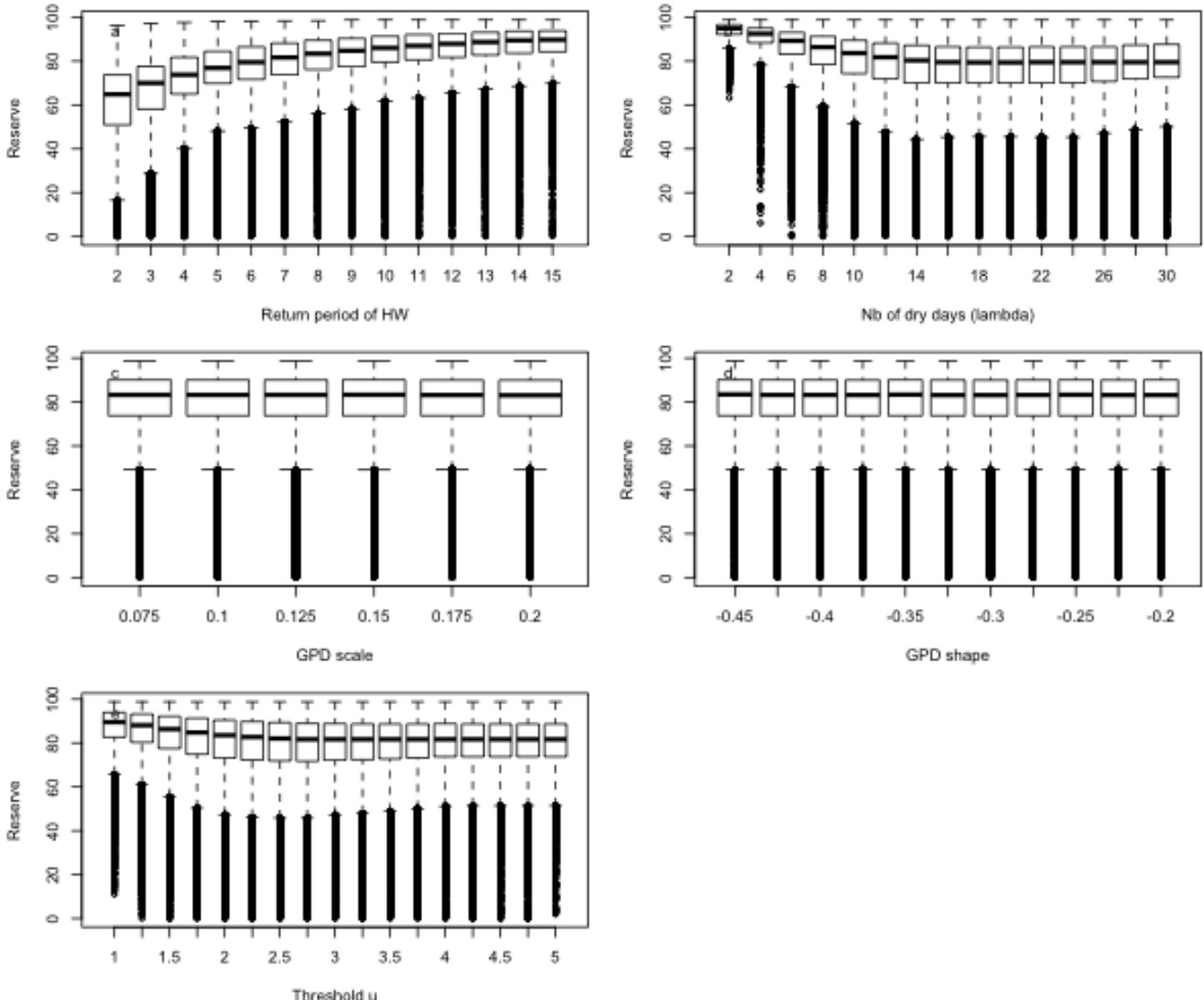

**Figure 3.** Dependence of probability distributions of average reserve before ruin $\langle R \rangle$ as a function of drought/heatwave (HW) return periods $\Lambda$ (a), number of hot days during summers $\lambda$ (b), GPD fit scale $\sigma$ (c) and GPD fit shape $\xi$ (d), and GPD threshold $u$ (e). Experiments are for $B = 0$. For each value of the control variable, a boxplot is given. The horizontal thick bar of boxplots represents the median ($q50$) of the distribution. The boxes boundaries represent the 25th quantile ($q25$) and the 75th quantile ($q75$). The upper whiskers are $\min[\max(\langle R \rangle), 1.5 \times (q75 - q25) + q50]$. The lower whisker has the symmetric formulation.

## 6 Conclusions

This paper presents a paradigm based on ruin theory to investigate tipping points for trees using a statistical framework. The framework illustrates how frequent occurrences of extreme events can damage trees to the point of dying. We described the

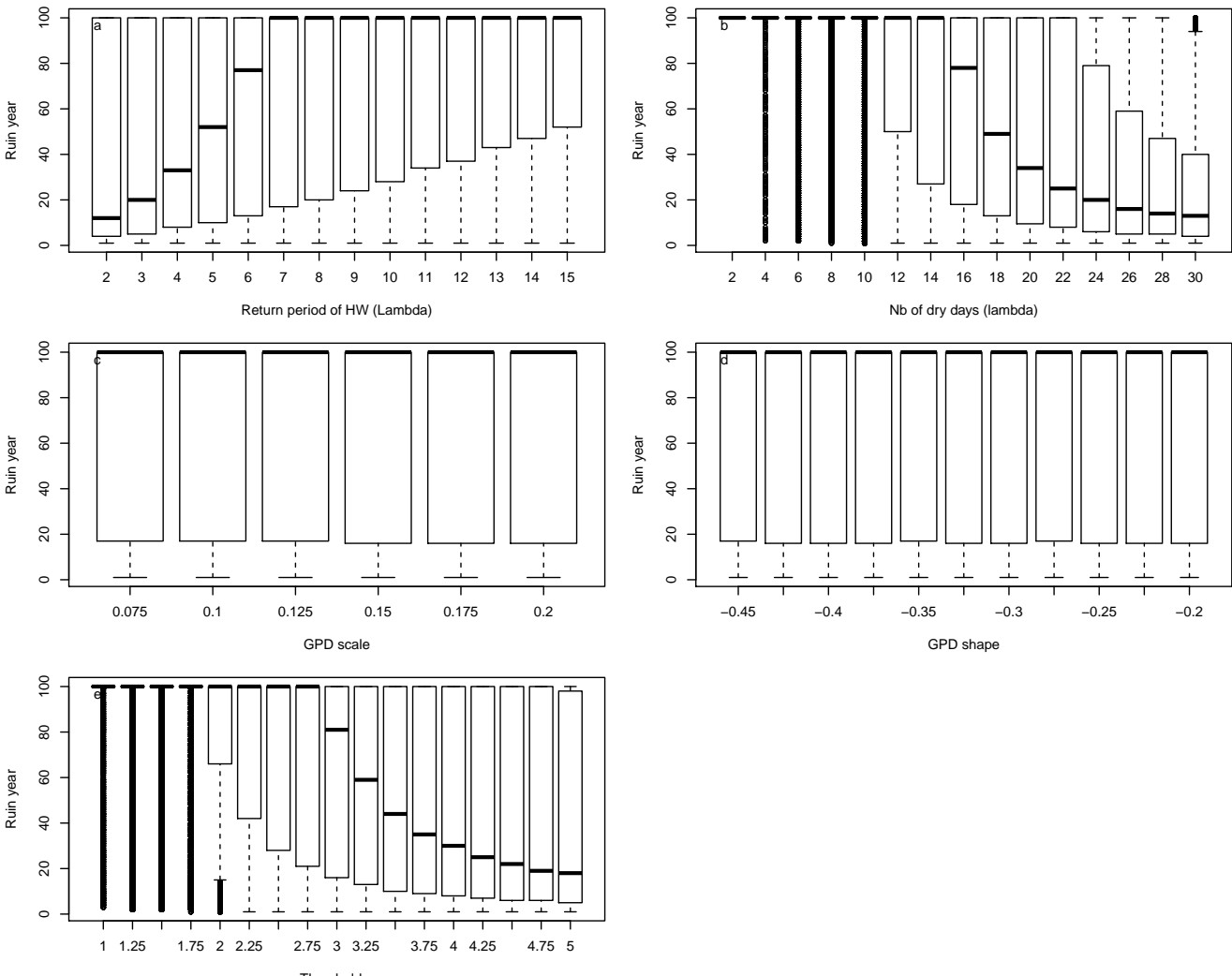

**Figure 4.** Dependence of probability distributions of ruin year $\tau$ as a function of drought/heatwave (HW) return periods $\Lambda$ (a), number of hot days during summers $\lambda$ (b), GPD fit scale $\sigma$ (c) and GPD fit shape $\xi$ (d), and threshold $u$ above which damage $S(t)$ is triggered (e). Experiments for $B = 0.4$. For each value of the control variable, a boxplot is given. The horizontal thick bar of boxplots represents the median ($q50$) of the distribution. The boxes boundaries represent the 25th quantile ($q25$) and the 75th quantile ($q75$). The upper whiskers are $\min[\max(\tau), 1.5 \times (q75 - q25) + q50]$. The lower whisker has the symmetric formulation. The points are for data that are above or below the whiskers.

vulnerability of an idealized forest by its probability of dying through the loss of non-structural carbohydrate reserves, and
350 illustrated a statistical methodology to identify climate parameters that control time to ruin. This proof of concept was applied to tree growth, but it could be extended to all types of eco-systems that are vulnerable to climate hazards on long time scales.

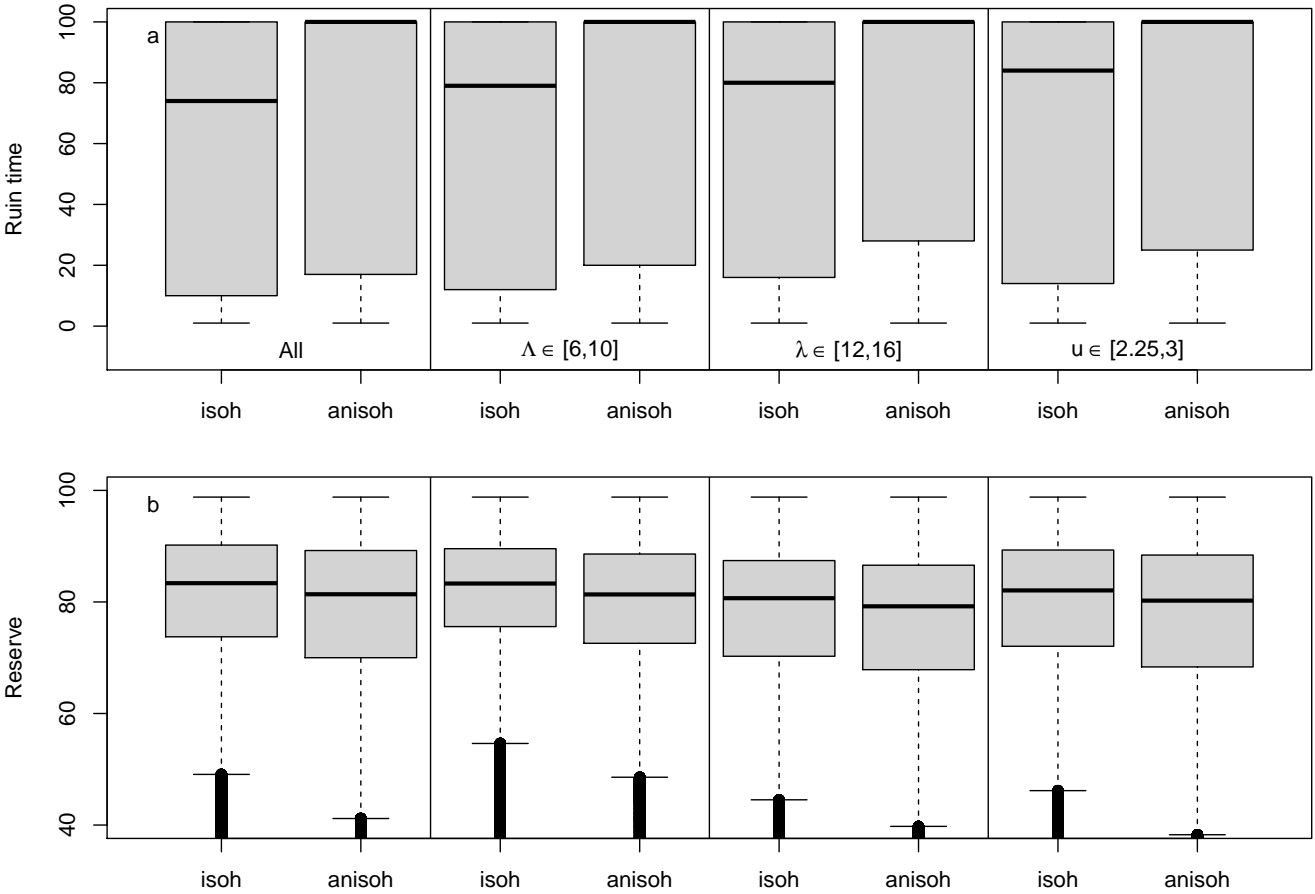

**Figure 5.** Conditional probability distribution of times to ruin (panel a) and reserve (panel b). The boxplots in the first column (isohydric (isoh: $B = 0$) and anisohydric (anisoh: $B = 0.4$)) are for all simulations. The boxplots in the second column are for return periods $\Lambda$ of droughts/heatwaves of 6–10 years for isohydric (isoh) and anisohydric (anisoh) simulations. The tboxplots in the third column are for durations $\lambda$ of 12–16 days of drought/heatwave in a summer. The boxplots in the last column are for GPD thresholds of 2.25–3. Note that points below the lower whiskers of the reserve boxplots are close to each other and appear as thick lines.

A careful determination of the physiological parameters of the model, from in situ observations, is necessary for an operational application of our approach.

The example shown here illustrates how forestry decisions may be guided from a priori information on climate change. The
framework shown here takes into account only one type of natural hazard. Others (including storms or fires) could be included, although the recovery rate (or "strategy") must be specified for each type of hazard.

We have investigated how variations of the hazard parameters affect the damage function and the probability of ruin. The most critical parameters are linked to the frequency of extreme events and their average intensity, which affect the rate of recovery of the trees.

We have examined the impact of "strategies" to cope with extremes. Although small, this impact has a differential effect on the average tree NSC reserve and the probability of ruin. For example, we obtain the counter-intuitive result that trees that have a lower probability of ruin on average also have a lower average reserve, if the hazard frequency changes (Figure 5). But if the hazard average intensity changes (longer or more intense), then trees with a lower probability of ruin also have a (slightly) higher reserve. Hence, changes in hazard intensity or frequency do not have the same effects on isohydric and anisohydric

coping strategies, although differences are small in our experiments.

This subtlety shows that this simple model can have different kinds of behavior as a response to hazards. In particular, this emphasizes the need to have a proper definition of the hazard (Cattiaux and Ribes, 2018), as this affects the Pareto or Poisson parameters of the hazard/damage function.

This study has obvious caveats. The growth/ruin model used here is exceedingly simple and does not reflect real trees,

as the Cramer-Lundberg does not reflect the complexity of the insurance sector. The proposed tree model is mainly a proof or concept, which could be improved by using parameters or processes with better physical and/or biological meaning. A key point of this work is that complex processes which are adversely affected by climatic conditions can be modelled using the Cramer-Lundberg modelling framework, which illustrates the competition between annual regular growth and occasional abrupt damages.

Some of the parameters (especially the impact scale factor $A_h$ in Eq. (9)) we used were chosen heuristically. Finer in situ studies and expertise would be necessary to tune those parameters for each tree species.

Many mathematical papers have described the exact properties of the Cramer-Lundberg model (e.g Embrechts et al., 1997, for a review). Our tree ruin/growth model violates some of the simple assumptions of the basic Cramer-Lundberg model, namely the time independence of $R(t)$ in the anisohydric mode. This forbids an analytic computation of times to ruin. This is

380 why we resort to extensive numerical Monte Carlo simulations. Those simulations ($\approx 10^6$ trajectories of 100 years) take less than 4 minutes on a 12 core computer.

The drought/heat stress index we have used is also rather crude and could be refined, although it was only designed to determine parameters of a Pareto distribution. We then simulated Pareto and Poisson distributions with parameters that are fitted to that index. All simulations were performed with the stationarity assumption (i.e. the parameters of the GPD distribution do

not change over time), albeit with random selections of parameters. Nonstationary simulations could be envisaged to explicitly take climate change into account. Data from climate model simulations could be used to simulate hazards (e.g. Herrera-Estrada and Sheffield, 2017; Massey et al., 2015). This is left to future studies.

*Code and data availability.* The simulation code and sample data to produce the drought index are available from https://zenodo.org/badge/ DOI/10.5281/zenodo.4075163.svg

## Appendix A: Drought/heatwave index definition

As this paper is more a proof-of-concept for a ruin model than a detailed study (which will be performed later), we consider simplified drought/heat hazard index that can easily be computed from climatological observations. We are interested in an index that reflects a compound event (Zscheischler et al., 2020) with extended dry period and high temperatures. There are a few indices of drought or aridity that consider precipitation and temperature (Baltas, 2007). The index of De Martonne (1926) normalizes cumulated precipitation and average temperature over a given season:

$$I_{DM} = \frac{P}{T+10},\tag{A1}$$

where the numerator is the cumulative precipitation (in mm/day) and $T$ is the mean temperature (in Celsius). This index was used to determine drought zones. Time variations of this index can be obtained by considering yearly or seasonal averages of precipitation and temperature. The $+10$ term in the denominator of Eq. (A1) is ad hoc to scale the respective variations of precipitation and temperature. Droughts are obtained with small values of this index (low precipitation and high temperature values).

Although easy to compute, this simple index suffers a few drawbacks. The main one is that it mainly reflects wetness, not drought. Most of of its variability is connected with the variability of precipitation and the upper tail of its probability distribution. Therefore seasons with little rain produce little variability in the index. One way to circumvent this would be to invert the index (i.e. consider $1/I_{DM}$). This is still not very satisfactory because the value of $I_{DM}$ for summers with notoriously dry heatwaves (e.g., 1976 and 2003) in Europe are just within the average and do not show anything special, contrary to what is expected (Ciais et al., 2005).

Thus, we propose an alternative drought index, still based on daily precipitation and temperature. For a given day $j$, we consider $D_j$ the frequency of precipitation: $D_j = 0$ if precipitation $P_j > 0.5$ mm/day, $D_j = 1$ if $P_j \leq 0.5$mm/day. We then construct an aridity index based on the weighted drought frequency and temperature.

$$I_{YV}(t) = \sum_{j=t-30}^{t} T_j \times (D_j + a) A \exp(-(t-j)/30)\tag{A2}$$

where $t$ is time (in days), $a \geq 0$ is a scaling constant (similar to $+10$ in Eq. (A1)) and $T_j$ is maximum daily temperature (TX in ECA&D (Klein-Tank et al., 2002)). $A \approx 30$ is a scaling constant to ensure that the sum of exponential weights is 1. In this paper, we chose $a = 1$ after a few tuning tests to verify that the index yields high values during notoriously dry years (e.g., 1976, 2003 or 2018).

This daily index is analogous to the inverse of the De Martonne index. One refinement comes from the exponential weights that give more importance to recent days than remote days. We can then compute the monthly median, upper quantiles and maximum of $I_{YV}$. We compute this index by starting on March 1st, as it is generally when the tree phenology resumes after the winter season, in the Northern midlatitudes. The daily index is computed until September 30th, when the vegetative cycle is almost finished.

With this new drought index, the extreme drought/heatwaves of 1976 and 2003 do become exceptional, as expected from the literature (Ciais et al., 2005). Figure A1 compares the precipitation, temperature, de Martonne and the new $I_{YV}$ for temperature and precipitation observations in Orly (near Paris, France). The precipitation or number of dry days do not yield extreme values for years with notorious heat stress in France (e.g. 1976, 2003 or 2019), as they are close to the 25–75th quantile values (Figure A1 panels b and d). Therefore, the De Martonne aridity index does not yield particularly extreme values for those years (Figure A1c).

To better evaluate how the new defined was a pertinent indicator for impact of climate on vegetation stress, we used the ORCHIDEE land surface model (Krinner et al., 2005) to simulate both the soil moisture and tree net primary production (NPP). We made a simulation using the ERA5 land atmospheric reanalyses at $0.1°×0.1°$ resolution (Hersbach et al., 2020) for the gridcell including Orly between 1981 and 2019. After a spin-up of 200 years using the first 10 year of the forcing, the simulation was done for the entire period. The variations of soil moisture serve as input for the hazard function and are a refinement over precipitation only. As an indicator of vegetation damage, we consider the number of days for which NPP is below the 10th quantile (0.15 gC/day/m$^2$), which indicates a "risk zone" for trees (Fig. A1f).

We find a significant (negative) correlation between the percentage of dry days and relative humidity for the 1981–2019 period (Fig. A1c), with $r = -0.7$ and $p$-value $< 10^{-6}$. The NPP variation are significantly (anti) correlated with the $I_{YV}$ ($r = -0.7$, $p$-value $< 10^{-6}$). The percentage of days for which NPP is below the 10th quantile corresponds to the exceedances of the $I_{YV}$ above a high threshold (Fig. A1e). Therefore, we believe that the choice of this index has a physical basis and can be used as a proxy to compute the parameters of a damage function.

We computed this $I_{YV}$ index on five stations from ECA&D (Klein-Tank et al., 2002): Berlin, Orly, Toulouse, De Bilt and Madrid. Those five stations cover a large part western Europe.

*Author contributions.* PY and NV co-constructed the growth model. PY designed the ruin model simulations. NV computed the relative humidity and NPP indices for validation. PY made the figures. Both authors contributed to the text.

*Competing interests.* The authors declare no competing interest.

*Disclaimer.* TEXT

*Acknowledgements.* This work was supported by an ERC grant No. 338965-A2C2 and a French ANR grant (SAMPRACE). We thank the two anonymous referees for their careful reviews that greatly helped improving the manuscript. We thank the Editor, Vivek Arora, who provided many additional comments that enhanced the clarity of the manuscript.

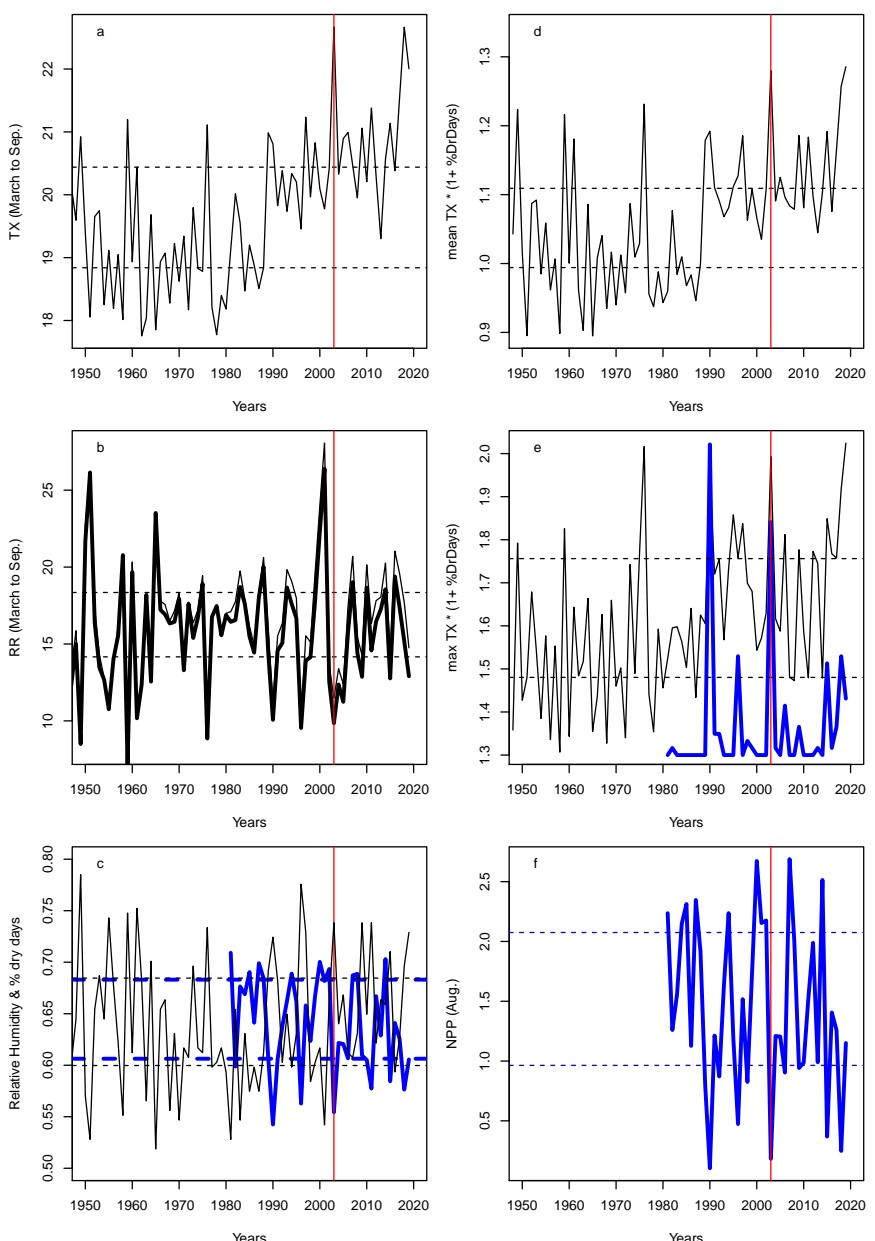

**Figure A1.** Variations of indices for Orly. Horizontal dashed lines for $q25$ and $q75$ quantiles. The vertical red lines are indicate the 2003 year. a: Average (March to September) temperature in Orly. b: Average (March to September) precipitation in Orly (thin line); scaled De Martonne (by 28) (thick black line). c: percentage of dry days between March and September (black line) and relative humidity (thick blue line). d: March to September mean of $I_{YV}$ . e: Daily maximum of $I_{YV}$ (black line) and scaled number of days when NPP is below the 10th quantile (thick blue line). f: NPP variations in Orly from an ORCHIDEE model simulation forced by the ERA5 reanalysis.

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
