# Peer review of "Modelling the Ruin of Forests under Climate Hazards"

_Earth System Dynamics, 2020_

## Referee Comment (RC1) · Anonymous Referee #1 · 5 Jan 2021

Review of Yiou

The authors adapt the 'ruin' theory of finance to that of forest growth. Using the base of the Cramer-Lundberg ruin model they develop a simple model to estimate tree survival/growth based on exposure to heatwaves/droughts. The work is interesting and novel but I struggled a bit with understanding what it is gaining over more conventional approaches to estimate ecosystem sensitivity to climate. I have three main issues with the paper. First, what is the true benefit of adopting this approach over, for e.g. a process-based model or one based on a simple statistical approach? This wasn't effectively conveyed in the paper. While the approach is obviously novel, it is needed to show that this is more than novelty for novelty's sake. Second, there is little attempt to evaluate if the model's outputs are reasonable. I think some attempt to evaluate the

results will be helpful. There was a small amount (e.g. L 180) but given the quantified thresholds and different behaviour between growth strategies, I think more could be done. Lastly, what exactly the model was predicting wasn't clear. At various points it was death of trees (presumably, L 52) or perhaps growth (L 181). They are, of course, linked but they are not the same thing. This conceptual muddiness makes for challenging reading. Lastly I would strongly suggest a toy problem to show how this model works on a known system. Since I think this paper is publishable, I suggest major revisions but it does need a large overhaul prior to that.

Specific comments:

Line 21 - What body of literature in the last few years? There is only one ref and it is from 2005.

L31 - I think 'ruin' needs to be properly defined. It is not a typical term in the papers of ESD and I am not yet sure at this point in the paper how I should interpret it.

L42 - Is this meant to imply that xylem embolism only kills branches and can't kill entire trees?

L45 - But trees typically carry far more carb reserves than needed to refoliate many times over, commonly dying with large reserves still intact. Also it is not clear whether carbon starvation is the leading cause of death in many cases (e.g. Rosas et al. 2013, Piper 2011, Rowland et al. 2015),

L50 - There needs to be a clear definition of 'ruin'. The farther I read the more convinced I am that this terminology needs to be more clearly set.

L52 - What is meant by 'disappearance of trees'? There death and total respiration?

L55 - what does 'average capital' mean here?

L59 - how is hazard defined in the context of this paper? It is a term that has many interpretations.

L65 - Please spend some time making this more intelligible to readers from the natural sciences. Pretty much nobody who reads ESD will come from a financial background so it is worth the word count to better expand on the terminology. E.g.'balance between competing companies' - what companies? 'the capital vanishes' - whose?

Eqn 1 - shouldn't pt have the t subscripted?

L71 - I suggest dropping the 'horizon' terminology. This might be a carryover from the insurance models but this is not a common way to talk about this in ecosystem modelling.

L79 - other insurance companies bidding for the same clients to sell insurance to?

- Small point, consider the tenses used. There are several instances of future tense that don't make sense. They make the reader wonder if this is going to appear in some future paper or ?

L102 - does this mean you don't let them allocate resources to their stems? So they can't grow?

L 104 - Fix the cite Handeregg.

L108 - does the p0 change spatially? How is it determined? Does B have an upper limit? Assumedly it is constrained such that p(t) >= 0 as I am not sure what a negative p would mean since the loss is supposed to come from the S term.

L118 - is there any ref you can use for proof of this here?

L155 - Can it be expanded upon how tho estimate these parameters from observations? It is one thing to suggest the possibility but I think it is more helpful to try and relate these parameters to something more grounded.

Section 2.2 - this whole section is a bit abstract. It would be beneficial to have included a toy example. Even a simple financial one where we could see how these equations play out. I would like to see that included especially with a figure. I think that would
benefit the paper and help the reader wrap their head around these concepts. After all, this is the first paper to intro the concept in the eco modelling lit.

L156 - So what units would all of these have? I am unsure what an Rmax means, 100%? If %, then of what?

L 161 - is this missing an "and" so would be 'and one of the trajectories with a ruin'?

L 164 - 'reserve units' - this seems like we should be able to use real units here.

Fig 1 a,b - please make it so that it is possible to get some info from this figure. Have multiple y-axis (sep plots) as right now it is not useful. Also I don't really understand how this works. Since it is 5-median-95 then I understand why the median is above the 5 and 95. But this looks like the actual 95 quantile realization was chosen (rather than the representative behaviour). Why not choose the mean and then give us the average behaviours? Right now it just looks like so much noise. As far as I can tell this figure is trying to make an important point that the model can capture differences due to aniso/iso strategies, so I think it is worth the effort to make it more convincing.

L177 - The model used here doesn't equate any reserves to stem (line 102). Is there any paper to point to that has directly linked the two? Does the Cailleret paper then do that? As written that isn't clear.

L181 - But aren't the simulations showing the decrease in reserves and not growth? If you are equating the reserves to growth, what does it mean when they trend to 0? No growth for a tree doesn't necessarily mean death but earlier that seems to be what is suggested (line 52). This whole paragraph is playing it very loose with terminology and relating poorly defined components of the ruin model with different real world observations. This needs to be tightened up considerably to be made consistent.

Fig 2 could be made into a table and my suggested toy example be added as a figure. Fig 2 has little interesting information that a table couldn't show.

Table 1 - HW = heatwave?

L239 - These numbers make me think you should then be able to go into the literature to find out how reasonable these are. Are there any reports that would substantiate what your model has found?

L241 - repeated text that makes it confusing.

Fig 4 - how is the avg reserve before ruin >0 when ruin was defined as R(t) = 0 (lines 70, 101)? How much before ruin is used in the calculation? I think this needs a time period defined and specified. Is 4e meant to have ruin year for the y axis label?

General - I would suggest that instead of 'cash' and 'credit' the terms be more ecological like 'aniso' and 'iso', it would help the reader place into context.

L 244 - Since the stat significance is mentioned here. Would it make sense to indicate in the figure which differences were significant?

L248 - I would not use 'globally' but rather something like 'On the whole'. Globally can be read as in, well, globally.

L256 - It does provide an estimate for sure, but I see little attempt here to evaluate the estimates. Can more effort be putting into evaluating the differences between the two strategies and whether any observations support the model results?

Refs cited:

Piper, F. I.: Drought induces opposite changes in the concentration of non-structural carbohydrates of two evergreen Nothofagus species of differential drought resistance, Ann. For. Sci., 68(2), 415–424, 2011.

Rosas, T., Galiano, L., Ogaya, R., Peñuelas, J. and Martínez-Vilalta, J.: Dynamics of non-structural carbohydrates in three Mediterranean woody species following long-term experimental drought, Front. Plant Sci., 4, 400, 2013.

Rowland, L., da Costa, A. C. L., Galbraith, D. R., Oliveira, R. S., Binks, O. J., Oliveira, A. A. R., Pullen, A. M., Doughty, C. E., Metcalfe, D. B., Vasconcelos, S. S., Ferreira, L. V.,

Malhi, Y., Grace, J., Mencuccini, M. and Meir, P.: Death from drought in tropical forests is triggered by hydraulics not carbon starvation, Nature, doi:10.1038/nature15539, 2015.

---

## Referee Comment (RC2) · Anonymous Referee #2 · 13 Jan 2021

Review of manuscript "Modelling the Ruin of Forests under Climate Hazards" submitted by Pascal Yiou and Nicolas Viovy

The submitted manuscript describes the application of the Cramer-Lundberg ruin model which is well-established in the insurance sector to tree mortality caused by droughts on 5 sites in Europe. It aims at introducing this model to climate and Earth system science to enable straightforward support for decision-making, something – as the authors claim – the tipping-points lacks. Because it is a simple model of tree mortality, tested at 5 climate stations in Europe, which describes the climate hazard events, I was wondering whether it would be more appropriate to transfer the manuscript to Natural Hazards and Earth System Science. In my view, the manuscript is lacking the feedback and resilience analysis and thus true interdisciplinary research to fit to the

scope of ESD. Furthermore, the manuscript is not well developed that it sets its new idea of applying the Cramer-Lundberg model to quantify tree mortality into the context of existing literature on modelling tree mortality due to drought (the climate hazard) under current and future climate change. It is hastily written and not sufficiently substantiated by the body of literature which is essential when introducing a new concept. I describe my major concerns in the following: 1) The introduction motivates the study with claims that a) the ecosystem service literature ignores the fact that ecosystem services are also threatened by disturbances or hazards, and b) tipping points are mostly qualitative, not providing probabilities, and policy makers make little use of such studies. Several problems arise with these claims. a. For a) the claim is simply not true, the ecosystem service literature does recognize climate extremes, incl. fires and drought, as disservices (see e.g. (Shackleton et al., 2016)). Further, the authors claim that it is a dogma that ecosystems provide services to society. I am not sure if the term "dogma" is a polemic claim or a misunderstanding from not translating it into a corresponding English term. The global IPBES assessment (Diaz et al., 2019) reflects the scientific agreement of an international body of scientists that this is the case. b. For b) Lenton et al. (2008) does provide the time scales at which the tipping points would occur and the literature on tipping points increasingly defines or refines those thresholds, e.g. Hirota et al. (2011) or Zemp et al. (2017) for the Amazon tipping element, or the Antarctic ice sheet (Garbe et al., 2020). Furthermore, it is not explained which limitations the tipping point concept has to answer the questions this paper aims to answer. 2) The introduction of collapsology to the Earth System Science community is not thoroughly done. One 15-year-old citation is provided in the introduction which is not sufficient to introduce the ESD readership to this scientific field which is unknown to this community. Again here, the state-of-the-art of this concept is not well described and the scientific gap not well developed. Furthermore, it is lacking a clear description of why a new concept is needed (things the ecosystem service concept cannot answer and the tipping point concept does not deliver), and why exactly this proposed concept is expected to provide a better solution. 3) The claims on the decision-making literature (from line 25)

[Figure]

is not supported by literature, so lacks evidence. The authors need to provide evidence or overview on how the tools established in insurance and finance provide "all the tools for decision-making", examples must be provided here to substantiate this claim. 4) The paper then later on does not get back to a tipping point/resilience or close collapse analysis nor does it make use of the ecosystems service-disservice concept. The authors do not get back to the issues raised in the introduction. This also applies to the decision-making tools mentioned in the introduction. 5) The general assumption is that the ruin of ecosystems can be captured with tree mortality. And tree mortality does not capture all patterns and processes of an ecosystem. This is an oversimplification that affects the outcome and interpretation of results of the study. Well, it only applies to wooded ecosystems. In addition to the description of forests affected by drought must be accompanied by an explanation on how the collapsology concept can be transformed to Earth system science, specifically ecosystem dynamics. This is the missing link which needs to be explained to correctly set the scene. An ecosystem is more complex than paying something in (GPP) and losing something (due to drought). So, the paper does not provide the evidence why the Cramer-Lundberg model or its extension is a better description of processes leading to drought-related tree mortality. 6) If the model has to produce 104 sample members, and it is shown for 5 meteorological stations only, I doubt its computational costs if applied to the global scale for a range of climate scenarios. 7) It is not explained why a new drought index had to be developed and why not existing and well-established drought indices could be used. This is important and missing in the manuscript. 8) Drought occurrence is not a random process. The assumption for $S(t)$ needs to be revised. Plants have more adaptation mechanisms by which they can avoid carbon starvation, loss of productivity (GPP) due to closed stomata and increased maintenance respiration. They have evolved physiological strategies and physiognomic structures to avoid transpiration loss. It can't be subsumed with having a carbon reserve pool or not. I can understand why this cannot be implemented in a simple model, but some notification of this knowledge is required to justify the model assumptions. 9) Lines 92-94: unclear how this can be

transformed to the tree-mortality application. This needs to be described here. Also, how this can help to advance science wrt drought impacts on increasing tree mortality and the stability of ecosystems. 10) Line 105, NPP needs to be properly introduced. Totally open, and not explained, how $p_0$ for the investment of NPP to the reserve pool can be justified. 11) Line 104: what is the damage function? $S(t)$ was introduced with a different meaning. 12) It needs to be shown that the climate data, i.e. number of droughts, indeed are Poisson and GPD distributions. 13) Line 155: the authors need to provide evidence that the parameter from their model can indeed be directly measured and evaluated using observations. This statement is not substantiated by evidence. 14) The findings that trees die at the time scale of decades to 100 years, is widely known and evidence is provided. The question is rather, if the model can produce increased drought-related mortality 3-5 years after a severe drought and the authors need to show how their findings compare to other model results or estimates based on drought-indices. There is an ample body of literature that has to be referenced here. Specifically, the result in line 182 indicates age mortality and not something related to a drought hazard. 15) Validation of modelled results is not provided and needs to be included.

References a. Diaz, S., Settele, J., Brondizio, E. S., Ngo, H. T., Agard, J., Arneth, A., Balvanera, P., Brauman, K. A., Butchart, S. H. M., Chan, K. M. A., Garibaldi, L. A., Ichii, K., Liu, J., Subramanian, S. M., Midgley, G. F., Miloslavich, P., Molnar, Z., Obura, D., Pfaff, A., Polasky, S., Purvis, A., Razzaque, J., Reyers, B., Chowdhury, R. R., Shin, Y. J., Visseren-Hamakers, I., Willis, K. J., and Zayas, C. N.: Pervasive human-driven decline of life on Earth points to the need for transformative change, Science, 366, 10.1126/science.aax3100, 2019. b. Garbe, J., Albrecht, T., Levermann, A., Donges, J. F., and Winkelmann, R.: The hysteresis of the Antarctic Ice Sheet, Nature, 585, 538-544, 10.1038/s41586-020-2727-5, 2020. c. Hirota, M., Holmgren, M., Van Nes, E. H., and Scheffer, M.: Global resilience of tropical forest and savanna to critical transitions, Science, 334, 232-235, 10.1126/science.1210657, 2011. d. Shackleton, C. M., Ruwanza, S., Sinasson Sanni, G. K., Bennett, S., De Lacy, P., Modipa,

R., Mtati, N., Sachikonye, M., and Thondhlana, G.: Unpacking Pandora's Box: Understanding and Categorising Ecosystem Disservices for Environmental Management and Human Wellbeing, Ecosystems, 19, 587-600, 10.1007/s10021-015-9952-z, 2016.

e. Zemp, D. C., Schleussner, C. F., Barbosa, H. M. J., and Rammig, A.: Deforestation effects on Amazon forest resilience, Geophysical Research Letters, 44, 6182-6190, 10.1002/2017gl072955, 2017.

---

## Author Comment (AC1) · 12 Feb 2021

Anonymous Review #1 We thank the reviewer for the constructive remarks that will help clarify the manuscript.

1. The authors adapt the 'ruin' theory of finance to that of forest growth. Using the base of the Cramer-Lundberg ruin model they develop a simple model to estimate tree survival/growth based on exposure to heatwaves/droughts. The work is interesting and novel but I struggled a bit with understanding what it is gaining over more conventional approaches to estimate ecosystem sensitivity to climate. I have three main issues with the paper. First, what is the true benefit of adopting this approach over, for e.g. a process-based model or one based on a simple statistical approach? This wasn't

effectively conveyed in the paper. While the approach is obviously novel, it is needed to show that this is more than novelty for novelty's sake.

The introduction will be rewritten. Our approach is a rather "simple statistical model", which explicitly focuses on the death of trees, and that is meant to explore the whole probability distributions of risks. The ruin model comes with the important concept of ensemble simulations to estimate probability distributions. Process based models yield computing limitations that hinder estimates of probability distributions. So, our proof of concept essentially paves the way for more extensive simulations with more sophisticated/realistic models.

2. Second, there is little attempt to evaluate if the model's outputs are reasonable. I think some attempt to evaluate the results will be helpful. There was a small amount (e.g. L 180) but given the quantified thresholds and different behaviour between growth strategies, I think more could be done.

Indeed, and this is acknowledged in the paper. Our argument is that if a complete mechanistic model can be simplified (e.g. through scaling analyses), our model presents the essential ingredients that it should contain. This will be clarified in the revised manuscript, especially in the discussion section. In addition, we will redo simulations with parameters that can be explicitly constrained by observations.

3. Lastly, what exactly the model was predicting wasn't clear. At various points it was death of trees (presumably, L 52) or perhaps growth (L 181). They are, of course, linked but they are not the same thing. This conceptual muddiness makes for chal-lenging reading.

As explained above, the objective of the model is to predict for which condition (both climate extreme events and the way vegetation deals with its capacity to manage such events) a whole forest will die by losing its capacity to grow. It is necessary for that to take into account for forest growth but this is not the final objective. This point will be clarified in the manuscript.

4. Lastly I would strongly suggest a toy problem to show how this model works on a known system. This comment is not clear to us: the model *is* a simplified model. Its toy version (i.e. without interannual memory) is the Cramer-Lundberg model, which is extensively described in the statistics literature.

Since I think this paper is publishable, I suggest major revisions but it does need a large overhaul prior to that.

Specific comments:

6. Line 21 - What body of literature in the last few years? There is only one ref and it is from 2005.

This point was overstated. We will tone down or remove the paragraph on collapsology, which might be far fetched, and which does not appear in the discussion.

7. L31 - I think 'ruin' needs to be properly defined. It is not a typical term in the papers of ESD and I am not yet sure at this point in the paper how I should interpret it.

The connection with ruin theory will be clarified in a more pedestrian way.

8. L42 - Is this meant to imply that xylem embolism only kills branches and can't kill entire trees?

No: xylem embolism can indeed kill the whole tree. But it can also only kill some parts of the tree which jeopardize its ability to growth during the following years. This will be corrected in the manuscript.

9. L45 - But trees typically carry far more carb reserves than needed to refoliate manytimes over, commonly dying with large reserves still intact. Also it is not clear whether carbon starvation is the leading cause of death in many cases (e.g. Rosas et al. 2013,Piper 2011, Rowland et al. 2015),

The reviewer is perfectly right: Trees in general die without a full depletion of carbohydrate reserves. However, a lot of studies, including those cited in the paper showed

that when trees die (not directly related to an extreme event as embolism) they are largely depleted in NSC compared to living ones. In particular as NSC are not only used for developing leaves but also for a lot of defending processes. So there is a level of NSC under which trees cannot survive to any supplementary stress. So in our representation a 0 value of R doesn't mean no NSC but a value under which the probability of tree die becomes very important. Likewise it is true that carbon starvation is one of the causes for tree death but not the main one: xylem embolism for instance is also an important cause. Partial embolism is also a factor that can alleviate the capability of trees to grow after extreme events. This is the reason why we mention the text embolism, and not only of carbon starvation. What is called "reserve" in the model should then be viewed not only as carbohydrate reserves but more generally "what can impact the growth of trees". We thank the reviewer for this comment, as this was confusing in the text. This will be better explained in the manuscript.

10. L50 - There needs to be a clear definition of 'ruin'. The farther I read the more convinced I am that this terminology needs to be more clearly set.

Thanks for the comment, as indeed the concept applied to forest needs to be clarified. Ruin in the context of the paper means that a forest stand cannot survive or, in other words the majority of trees are dying. This concept of ruin can be applied to a forest in general but also to a given tree species, which means in this case that this species will disappear from the forest but that a forest with other more tolerant species can still exist. The definition will be clarified in the revised manuscript

11. L52 - What is meant by 'disappearance of trees'? There death and total respiration?

Thanks for the comment. The term was not clear: this just means tree death. It will be corrected

12. L55 - what does 'average capital' mean here?

It means "average reserve". This will be added.

13. L59 - how is hazard defined in the context of this paper? It is a term that has many interpretations.

"Hazard" means: a potential source of harm (see e.g. https://en.wikipedia.org/wiki/Hazard). Our modelling exercise illustrates the three terms of that definition (potential, source, and harm). This will be clarified in the text.

14. L65 - Please spend some time making this more intelligible to readers from the natural sciences. Pretty much nobody who reads ESD will come from a financial background so it is worth the word count to better expand on the terminology. E.g.'balance between competing companies' - what companies? 'the capital vanishes' - whose?

OK. This will be clarified (see point L79 below).

15. Eqn 1 - shouldn't pt have the t subscripted?

No: it is p \times t. A fixed premium rate p is collected each year t. Of course, p could also depend on t (which is what we do in Eq. (6)). We will add a $\cdot$ between p and t in Eq. (1) to clarify this.

16. L71 - I suggest dropping the 'horizon' terminology. This might be a carry-over fromthe insurance models but this is not a common way to talk about this in ecosystemmodelling. The summary for policymakers of the IPCC AR5 (1st chapter; https://www.ipcc.ch/srocc/chapter/summary-for-policymakers/) states:

"C.1.1 The temporal scales of climate change impacts in ocean and cryosphere and their societal consequences operate on time horizons which are longer than those of governance arrangements (e.g., planning cycles, public and corporate decision making cycles, and financial instruments)."

We use "horizon" with the same meaning of the IPCC AR5. Therefore, we think that it is appropriate to use "horizon" when speaking of a time bound in an ESD paper. We will make this clear in the manuscript, so that the readers who are not familiar with IPCC language do not think that it is insurance jargon, which we do not speak.

17. L79 - other insurance companies bidding for the same clients to sell insurance to?- Small point, consider the tenses used. There are several instances of future tense that don't make sense. They make the reader wonder if this is going to appear in some future paper or ?

We meant to say that, as insurance companies compete with each other to get clients (i.e., you or us), they have to find a balance between a high premium rate (to ensure an income for them) and attracting clients from other companies (i.e. being somewhat cheaper). The necessity for this economic balance gives an upper bound for the premium rate. In other words, having a very high premium rate might not solve the problem of ruin for insurance companies, because they would lose all their clients to less greedy companies. This basic concept of capitalist economy will be clarified in the text, although it is marginally important for the paper. The future tenses will be checked.

18. L102 - does this mean you don't let them allocate resources to their stems? So they can't grow?

No: obviously it means allocation to all the organs of the plants, so we will correct this.

19. L 104 - Fix the cite Handeregg.

OK. Sorry for the typo.

20. L108 - does the p0 change spatially?

Here we considered the model on a given point, and in the simplified application presented in the paper we assume a constant NPP, but obviously if this model is applied regionally p0 can change spatially (as well as the others model parameters).

21. How is it determined?

p0 is heuristically chosen so that Rmax is reached in 20 years when no hazard occurs. This choice is debatable and could be constrained by tree species. Following

the two reviewers' remarks we will revise the different parameters (including p0) to be defined from literature and then better represent real cases. This will be clarified in the manuscript.

22. Does B have an upper limit? Assumedly it is constrained such that p(t) >= 0 as I am not sure what a negative p would mean since the loss is supposed to come from the S term.

Indeed: if trees do not shrink (negative p), an upper bound for B would be $p_0/E(S)$, where $E(S)$ is the mathematical expectation of $S$.

23. L118 - is there any ref you can use for proof of this here?

This is a classical result in statistics: the probability distribution of the number of times that a time series exceeds a high threshold converges to a Poisson distribution. This will be clarified in the text, with a reference to classical but pedestrian textbooks (e.g. Coles, S. An introduction to statistical modeling of extreme values. Springer series in statistics. London, New York: Springer, 2001.), and references in the atmospheric science literature (e.g. Smith, R. L., and T. S. Shively. Point process approach to modeling trends in tropospheric ozone based on exceedances of a high threshold. Atmospheric Environment 29(1995): 3489 99.)

24. L155 - Can it be expanded upon how to estimate these parameters from observations? It is one thing to suggest the possibility but I think it is more helpful to try and relate these parameters to something more grounded.

Our sentence is indeed vague. $A_h$ and $B$ relate how climate hazards impact the growth of trees. As a first approximation, they could be obtained from a correlation between the climate conditions during identified droughts (e.g. 1976, 2003, 2018, 2019) and tree growth parameters (e.g. tree ring width/density, NPP), which are determined in situ of from satellite observations (NPP). We believe that obtaining such a correlation/regression is a whole methodological paper in itself. This important point will be

discussed in the revised manuscript.

25. Section 2.2 - this whole section is a bit abstract. It would be beneficial to have included a toy example. Even a simple financial one where we could see how these equations play out. I would like to see that included especially with a figure. I think that would benefit the paper and help the reader wrap their head around these concepts. After all,this is the first paper to intro the concept in the eco modelling lit.

26. Section 2.3 (sample trajectories) just does that by illustrating what trajectories look like.

27. L156 - So what units would all of these have? I am unsure what an Rmax means,100%? If %, then of what?

A percentage could be more meaningful, although this would not change the gist of the paper (see point below). This will be changed in the revised manuscript.

28. L 164 - 'reserve units' - this seems like we should be able to use real units here.

Rmax represents the upper limit of tree "reserves" as tree do not tend to accumulate reserves indefinitely. If a sufficient level is reached plant will allocate its remaining assimilate to growth of different organs. Concerning the units, if we considered that R represents stricto sensu carbohydrate reserve we should indeed use gC.m2 for instance. However as discussed previously and noticed by the reviewer, considering only carbohydrate reserve as driver of tree growth and tree die is too restrictive as others processes as xylem embolism should also be considered. So in such a theoretical model, "reserve" should be considered as all "that allow trees to initiate growth on next year" and then cannot be associated to a specific unit and then 100 is an arbitrary unit scaled with the others parameters of the model and calibrated to give reasonable results compared to observations.

29. L 161 - is this missing an "and" so would be 'and one of the trajectories with a ruin'?

OK. An "and" will be added.

30. Fig 1 a,b - please make it so that it is possible to get some info from this figure. Have multiple y-axis (sep plots) as right now it is not useful. Also I don't really understand how this works. Since it is 5-median-95 then I understand why the median is above the 5 and 95. But this looks like the actual 95 quantile realization was chosen (rather than the representative behaviour). Why not choose the mean and then give us the average behaviours? Right now it just looks like so much noise. As far as I can tell this figure is trying to make an important point that the model can capture differences due to aniso/iso strategies, so I think it is worth the effort to make it more convincing.

Sorry: there is typo. The 95th quantile should be in blue and the median in black. The caption will be corrected. The upper panels are indeed hard to read. Another representation that synthesizes the statistical features of the identified trajectories will be proposed.

31. L177 - The model used here doesn't equate any reserves to stem (line 102). Is there any paper to point to that has directly linked the two? Does the Cailleret paper then do that? As written that isn't clear.

In the text we mention only allocation to roots and leaves because these are the organs that play a direct role in growing. But obviously it does mean that reserves are not allocated to other parts of the plant as stems and trunk. But to avoid confusion will we correct this in the text.

32. L181 - But aren't the simulations showing the decrease in reserves and not growth?If you are equating the reserves to growth, what does it mean when they trend to 0? No growth for a tree doesn't necessarily mean death but earlier that seems to be what is suggested (line 52). This whole paragraph is playing it very loose with terminology and relating poorly defined components of the ruin model with different real world observations. This needs to be tightened up considerably to be made consistent.

Thanks for the comment. Actually, a living tree cannot have null growth anyway. However, Indeed growth is obviously different from reserves. In particular with the same level of NSC the tree ring growth will depend on the climate condition of the year. But considering the relative difference between tree ring growth of healthy trees and dying trees is used as a proxy of NSC in papers like Caillerets et al. since it allows to disentangle the effect of climate and then it is well correlated to NSC status (as it is not possible to directly measure NSC of trees already dead). In the model, we also consider a relative change in reserve status (as mentioned before Rmax is fixed to an arbitrary value of 100), the comparison between the change in R and the relative change in tree ring growth make qualitatively sense. This will be detailed in the text.

33. Fig 2 could be made into a table and my suggested toy example be added as a figure.Fig 2 has little interesting information that a table couldn't show. OK. This figure was meant to reassure statisticians, but Table 1 summarizes the same information.

34. Table 1 - HW = heatwave?

Yes. This will be clarified in the revised manuscript.

35. L239 - These numbers make me think you should then be able to go into the literature to find out how reasonable these are. Are there any reports that would substantiate what your model has found?

This type of estimate is coherent with other studies (e.g. Parey et al. Validation of a stochastic temperature generator focusing on extremes, and an example of use for climate change. Climate Research 59, no 1 (2014): 61 75; Kharin et al. Changes in temperature and precipitation extremes in the CMIP5 ensemble. Climatic change 119, no 2 (2013): 345 57).

36. L241 - repeated text that makes it confusing.

This will be corrected to avoid confusion.

37. Fig 4 - how is the avg reserve before ruin >0 when ruin was defined as R(t) = 0

(lines70, 101)? How much before ruin is used in the calculation? I think this needs a time period defined and specified. Is 4e meant to have ruin year for the y axis label?General - I would suggest that instead of 'cash' and 'credit' the terms be more ecologi-cal like 'aniso' and 'iso', it would help the reader place into context.

We show the average of reserve conditional to time before ruin. So, if $\tilde\tau$ is the min between the ruin time and 100 years, we show the distribution (along all samples) of $1/{\tilde\tau} \sum_{t=1}^{t=\tilde\tau} R(t)$. This will be clarified in the text. We will rephrase the terminology to a more ecological context.

38. L 244 - Since the stat significance is mentioned here. Would it make sense to indicate in the figure which differences were significant?

This will be emphasized in the revised manuscript.

39. L248 - I would not use 'globally' but rather something like 'On the whole'. Globally can be read as in, well, globally.

OK.

40. L256 - It does provide an estimate for sure, but I see little attempt here to evaluate the estimates. Can more effort be putting into evaluating the differences between the two strategies and whether any observations support the model results?

OK. We will provide examples of simulations with values that are constrained by the literature (e.g., He W, et al., Patterns in nonstructural carbohydrate contents at the tree organ level in response to drought duration. Glob Chang Biol. 2020 Jun;26(6):3627-3638. doi: 10.1111/gcb.15078).

Refs cited: Piper, F. I.: Drought induces opposite changes in the concentration of non-structuralcarbohydrates of two evergreen Nothofagus species of differential drought resistance,Ann. For. Sci., 68(2), 415–424, 2011. Rosas, T., Galiano, L., Ogaya, R., Peñuelas, J. and Martínez-Vilalta, J.: Dynamics of non-structural carbohydrates in three Mediterranean woody species following long-term experimental drought, Front.

Plant Sci., 4, 400, 2013. Rowland, L., da Costa, A. C. L., Galbraith, D. R., Oliveira, R. S., Binks, O. J., Oliveira, A.A. R., Pullen, A. M., Doughty, C. E., Metcalfe, D. B., Vasconcelos, S. S., Ferreira, L. V.,C5Malhi, Y., Grace, J., Mencuccini, M. and Meir, P.: Death from drought in tropical forests is triggered by hydraulics not carbon starvation, Nature, doi:10.1038/nature15539,2015

---

## Author Comment (AC2) · 12 Feb 2021

Anonymous Review #2

We thank the referee for the efforts devoted to this review.

The submitted manuscript describes the application of the Cramer-Lundberg ruin model which is well-established in the insurance sector to tree mortality caused by droughts on 5 sites in Europe. It aims at introducing this model to climate and Earth system science to enable straightforward support for decision-making, something – as the authors claim – the tipping-points lacks. Because it is a simple model of tree mortality, tested at 5 climate stations in Europe, which describes the climate hazard events,

0. I was wondering whether it would be more appropriate to transfer the manuscript to Natural Hazards and Earth System Science. In my view, the manuscript is lacking the feedback and resilience analysis and thus true interdisciplinary research to fit to the scope of ESD. Furthermore, the manuscript is not well developed that it sets its new idea of applying the Cramer-Lundberg model to quantify tree mortality into the context of existing literature on modelling tree mortality due to drought (the climate hazard) under current and future climate change. It is hastily written and not sufficiently substantiated by the body of literature which is essential when introducing a new concept.

We do have a different view of interdisciplinarity (in this paper: putting together ideas from econometrics and earth sciences). This comment from the reviewer on the relevance to the scope of ESD seems based on his/her personal feeling. Our manuscript seems more interdisciplinary than most recently published papers in ESD, including from the editors-in-chief of the journal.

I describe my major concerns in the following:

1) The introduction motivates the study with claims that a) the ecosystem service literature ignores the fact that ecosystem services are also threatened by disturbances or hazards,

We do not understand this point raised by the reviewer as we never claimed that ecosystem service literature ignores disturbances and hazards. Moreover, our paper is not about ecosystem services.

and b) tipping points are mostly qualitative, not providing probabilities, and policy makers make little use of such studies. Several problems arise with these claims. a. For a) the claim is simply not true, the ecosystem service literature does recognize climate extremes, incl. fires and drought, as disservices (see e.g. (Shackleton et al., 2016)). Further, the authors claim that it is a dogma that ecosystems provide services to society. I am not sure if the term "dogma" is a polemic claim or a misunderstanding from not translating it into a corresponding English term. The global IPBES assessment (Diaz

et al., 2019) reflects the scientific agreement of an international body of scientists that this is the case. b. For b) Lenton et al. (2008) does provide the time scales at which the tipping points would occur and the literature on tipping points increasingly defines or refines those thresholds, e.g. Hirota et al. (2011) or Zemp et al. (2017) for the Amazon tipping element, or the Antarctic ice sheet (Garbe et al., 2020). Furthermore, it is not explained which limitations the tipping point concept has to answer the questions this paper aims to answer.

We will rephrase the introduction. As stated in the title, we focus on trees, not ice sheets. We opted to cite seminal papers, which contain the main ideas, while the newer ones are essentially applications of existing tools and concepts.

2) The introduction of collapsology to the Earth System Science community is not thoroughly done. One 15-year-old citation is provided in the introduction which is not sufficient to introduce the ESD readership to this scientific field which is unknown to this community. Again here, the state-of-the-art of this concept is not well described and the scientific gap not well developed. Furthermore, it is lacking a clear description of why a new concept is needed (things the ecosystem service concept cannot answer and the tipping point concept does not deliver), and why exactly this proposed concept is expected to provide a better solution.

We did not mean to write an introduction to collapsology, but merely related our study to that concept. Citing the seminal piece of work, which contains all elements of understanding, seems more efficient than citing recent papers, which are based on the original one. The introduction will be rephrased in order to remove the discussion on collapsology, which can sound far fetched.

3) The claims on the decision-making literature (from line 25) is not supported by literature, so lacks evidence. The authors need to provide evidence or overview on how the tools established in insurance and finance provide "all the tools for decision-making", examples must be provided here to substantiate this claim.

The reviewer seems to overinterpret our claim. We obviously do not claim that the insurance sector provides all tools for all decisions: we claim that the insurance sector has developed all tools that are necessary to forecast its own losses and benefits. Such tools seem to be accepted by a majority of humans who are willing to pay a fee to insure their health, house or car. We believe that this fact speaks for itself.

4) The paper then later on does not get back to a tipping point/resilience or close collapse analysis nor does it make use of the ecosystems service-disservice concept. The authors do not get back to the issues raised in the introduction. This also applies to the decision-making tools mentioned in the introduction.

Indeed. This is why the introduction will be rephrased. Thank you for those comments. The paper is not about ecosystem service or disservice.

5) The general assumption is that the ruin of ecosystems can be captured with tree mortality. And tree mortality does not capture all patterns and processes of an ecosystem. This is an oversimplification that affects the outcome and interpretation of results of the study. Well, it only applies to wooded ecosystems. In addition to the description of forests affected by drought must be accompanied by an explanation on how the collapsology concept can be transformed to Earth system science, specifically ecosystem dynamics. This is the missing link which needs to be explained to correctly set the scene. An ecosystem is more complex than paying something in (GPP) and losing something (due to drought). So, the paper does not provide the evidence why the Cramer-Lundberg model or its extension is a better description of processes leading to drought-related tree mortality.

We never claim that the ruin of ecosystems (in general) stems from tree mortality. We focus on tree mortality (which is called "ruin"), then claim that this concept of ruin modelling can be transposed to other ecosystems, which would obviously require a specific model adjustment.

For example, the paper does not address the question of tipping points in herbaceous

ecosystems which have a very different behavior in response to climate hazards. We will make this more explicit in the revised manuscript.

6) If the model has to produce 104 sample members, and it is shown for 5 meteorological stations only, I doubt its computational costs if applied to the global scale for a range of climate scenarios.

Indeed, reconstructing a probability distribution requires producing a huge number of simulations, which would not be possible with a complex process-based model, but which is not a problem with the ruin model as obtaining the Generalized Pareto Distribution parameters of any variable at a global scale is a matter of minutes on a PC, and a few seconds on a parallel computer (e.g., Kharin, V.V et al. Changes in temperature and precipitation extremes in the CMIP5 ensemble. Climatic change 119, no 2 (2013): 345‑57), and is done only once. Doing our 10000 Monte-Carlo simulations from those parameters for Europe takes a couple of minutes on a PC. Doing it at a global scale would take a few hours on a PC (at most), a few minutes on a supercomputer, as computations can be parallelized.

7) It is not explained why a new drought index had to be developed and why not existing and well-established drought indices could be used. This is important and missing in the manuscript.

A new drought index is indeed marginally important for the ruin model, which is why it was deferred to an appendix. What is important in the ruin model are the parameters of the probability distribution of hazards (intensity, duration and frequency). Well-established drought indices (such as the one of de Martone) are not physically satisfactory, as explained in the appendix. There are other more physically satisfactory drought indices, which do account for physical processes, but they do not go that far into the past and do not allow a reliable estimate of parameters for the hazard model. This motivated this new drought index, which can be computed with basic climatic variables that are recorded over long periods of time.

8) Drought occurrence is not a random process. The assumption for S(t) needs to be revised. Plants have more adaptation mechanisms by which they can avoid carbon starvation, loss of productivity (GPP) due to closed stomata and increased maintenance respiration. They have evolved physiological strategies and physiognomic structures to avoid transpiration loss. It can't be subsumed with having a carbon reserve pool or not. I can understand why this cannot be implemented in a simple model, but some notification of this knowledge is required to justify the model assumptions.

We do not agree with the preamble of this comment. The phenomenological development of a drought (or any climate hazard) is obviously a deterministic process, but key quantities like frequency, duration or intensity can be modeled by random processes. This is the core of statistical climatology.

9) Lines 92-94: unclear how this can be transformed to the tree-mortality application. This needs to be described here. Also, how this can help to advance science wrt drought impacts on increasing tree mortality and the stability of ecosystems.

Our claim (in those lines 92-94) is that it is possible to determine the probability distribution of ruin time from statistical properties of hazards. The following section (2.2) shows how the Cramer-Lundberg model (and the surrounding probabilistic framework) can be transposed to tree-mortality (that we generically call "ruin"). This will be made more explicit in the text.

10) Line 105, NPP needs to be properly introduced. Totally open, and not explained, how p_0 for the investment of NPP to the reserve pool can be justified.

An NPP definition will be added. It has been shown that in good condition, the non structural carbohydrate reserves in mature trees tend to reach a maximum value and then bad weather conditions decrease this amount both by increasing use of reserves and decreasing allocation to it (because of reduced total NPP). (Barbaroux, C., Bréda, N., Dufrêne, E., 2003. Distribution of aboveground and belowground carbohydrate reserves inadult trees of two contrasting broad-leaved species (QuercuspetraeaandFagus sylvatica). New Phytol. 157, 605–615. This justifies the assumption of the model of a maximum

11) Line 104: what is the damage function? S(t) was introduced with a different meaning.

S(t) is a "hazard function". We will streamline the terminology between sections 2.1 and 2.2.

12) It needs to be shown that the climate data, i.e. number of droughts, indeed are Poisson and GPD distributions.

Under rather generic mathematical conditions, the probability distribution of the exceedances over a threshold can be modeled by a Generalized Pareto Distribution. This is analogous to the fact that the average of a variable that has a finite standard deviation can be modeled by a Gaussian. The principle is the same for the frequency of exceedances (or inter-arrival times). The simplest statistical model that describes the time interval between exceedances of a threshold is a Poisson distribution. We will recall this fact in the text by citing textbooks that are often used in statistical climatology or hazard models (e.g. S. Coles, An Introduction to Statistical Modeling of Extreme Values, Springer, 2001).

13) Line 155: the authors need to provide evidence that the parameter from their model can indeed be directly measured and evaluated using observations. This statement is not substantiated by evidence.

The parameters were initially chosen arbitrarily, in order to illustrate the contrasting behavior of "cash" and "credit" strategies. This has been mentioned in the discussion of the paper. We will re-do analyses that use constraints obtained from the literature, e.g. He W, et al. (Patterns in nonstructural carbohydrate contents at the tree organ level in response to drought duration. Glob Chang Biol. 2020 Jun;26(6):3627-3638. doi: 10.1111/gcb.15078), who performed a meta-analysis of reserve reduction in case

of severe drought.

14) The findings that trees die at the time scale of decades to 100 years, is widely known and evidence is provided. The question is rather, if the model can produce increased drought-related mortality 3-5 years after a severe drought and the authors need to show how their findings compare to other model results or estimates based on drought-indices. There is an ample body of literature that has to be referenced here. Specifically, the result in line 182 indicates age mortality and not something related to a drought hazard.

Thank you for this comment. Indeed, very few trees in Europe live longer than a century. We consider that our study applies to a collection of trees (e.g. a forest), whose lifetime is hoped to be longer than a century. This will be clarified in the text.

15) Validation of modelled results is not provided and needs to be included.

Although we appreciate this comment (also made by referee#1), we would like to point out that since the seminal paper of E.N. Lorenz (Deterministic nonperiodic flow. J. Atmos. Sci. 20 (1963): 130‑41), many studies, including publications in ESD, have focused on the behavior of idealized models.

References a. Diaz, S., Settele, J., Brondizio, E. S., Ngo, H. T., Agard, J., Arneth, A., Balvanera, P., Brauman, K. A., Butchart, S. H. M., Chan, K. M. A., Garibaldi, L. A., Ichii, K., Liu, J., Subramanian, S. M., Midgley, G. F., Miloslavich, P., Molnar, Z., Obura, D., Pfaff, A., Polasky, S., Purvis, A., Razzaque, J., Reyers, B., Chowdhury, R. R., Shin, Y. J., Visseren-Hamakers, I., Willis, K. J., and Zayas, C. N.: Pervasive human-driven decline of life on Earth points to the need for transformative change, Science, 366, 10.1126/science.aax3100, 2019. b. Garbe, J., Albrecht, T., Levermann, A., Donges, J. F., and Winkelmann, R.: The hysteresis of the Antarctic Ice Sheet, Nature, 585, 538-544, 10.1038/s41586-020-2727-5, 2020. c. Hirota, M., Holmgren, M., Van Nes, E. H., and Scheffer, M.: Global resilience of tropical forest and savanna to critical transitions, Science, 334, 232-235, 10.1126/science.1210657, 2011. d. Shack-

leton, C. M., Ruwanza, S., Sinasson Sanni, G. K., Bennett, S., De Lacy, P., Modipa, R., Mtati, N., Sachikonye, M., and Thondhlana, G.: Unpacking Pandora's Box: Understanding and Categorising Ecosystem Disservices for Environmental Management and Human Wellbeing, Ecosystems, 19, 587-600, 10.1007/s10021-015-9952-z, 2016. e. Zemp, D. C., Schleussner, C. F., Barbosa, H. M. J., and Rammig, A.: Deforestation effects on Amazon forest resilience, Geophysical Research Letters, 44, 6182-6190, 10.1002/2017gl072955, 2017.

We thank the referee for those interesting recent references.

————————————————————

---

## Author Response (AR1)

**Anonymous Review #1**

The reviewer's comments are in red, our replies are in blue. We thank the reviewer for the constructive remarks that help clarifying the manuscript.

The authors adapt the 'ruin' theory of finance to that of forest growth. Using the base of the Cramer-Lundberg ruin model they develop a simple model to estimate tree sur-vival/growth based on exposure to heatwaves/droughts. The work is interesting and novel but I struggled a bit with understanding what it is gaining over more conventional approaches to estimate ecosystem sensitivity to climate. I have three main issues with the paper.
First, what is the true benefit of adopting this approach over, for e.g. a process-based model or one based on a simple statistical approach? This wasn't effectively conveyed in the paper. While the approach is obviously novel, it is needed to show that this is more than novelty for novelty's sake.

The introduction was rewritten (i.e. simplified and more focused). Our approach is a rather "simple statistical model", which explicitly focuses on the death of trees, and that is meant to explore the whole probability distributions of risks. The ruin model comes with the important concept of ensemble simulations to estimate probability distributions. Process-based models yield computing limitations that hinder estimates of probability distributions. So, our proof of concept essentially paves the way for more extensive simulations with more sophisticated/realistic models.

Second, there is little attempt to evaluate if the model's outputs are reasonable. I think some attempt to evaluate the results will be helpful. There was a small amount (e.g. L 180) but given the quantified thresholds and different behaviour between growth strategies, I think more could be done.

Indeed, and this is acknowledged in the paper (end of section 2.3). Our argument is that if a complete mechanistic model can be simplified (e.g. through scaling analyses), our model presents the essential ingredients that it should contain. This will be clarified in the revised manuscript, especially in the discussion section. In addition, we redid simulations with parameters that can be explicitly constrained by observations.

Lastly, what exactly the model was predicting wasn't clear. At various points it was death of trees (presumably, L 52) or perhaps growth (L 181). They are, of course,linked but they are not the same thing. This conceptual muddiness makes for challenging reading.

As explained above, the objective of the model is to evaluate for which condition (both climate extreme events and the way trees deal with their capacity to manage such

events) a whole forest can die by losing its capacity to grow. It is necessary for that to take into account for forest growth but this is not the final objective. This point is clarified in the manuscript, especially in the revised introduction.

Lastly I would strongly suggest a toy problem to show how this model works on a known system.
This comment is not clear to us: the model *is* a simplified model. Its toy version (i.e. without interannual memory) is the Cramer-Lundberg model, which is extensively described in the statistics literature.

Since I think this paper is publishable, I suggest major revisions but it does need a large overhaul prior to that.

**Specific comments:**

Line 21 - What body of literature in the last few years? There is only one ref and it is from 2005.
This point was overstated. We will tone down or remove the paragraph on collapsology, which might be far fetched, and which does not appear in the discussion.

L31 - I think 'ruin' needs to be properly defined. It is not a typical term in the papers of ESD and I am not yet sure at this point in the paper how I should interpret it.
The connection with (statistical) ruin theory will be clarified in a more pedestrian way. We now state explicitly what is meant by ruin in the formulation of the model.

L42 - Is this meant to imply that xylem embolism only kills branches and can't kill entire trees?
No: xylem embolism can indeed kill the whole tree. But it can also only kill some parts of the trees, which jeopardizes their ability to grow during the following years. This is corrected in the manuscript.

L45 - But trees typically carry far more carb reserves than needed to refoliate many times over, commonly dying with large reserves still intact. Also it is not clear whether carbon starvation is the leading cause of death in many cases (e.g. Rosas et al. 2013,Piper 2011, Rowland et al. 2015),

The reviewer is perfectly right: in general, trees die without a full depletion of carbohydrate reserves. However, a lot of studies, including those cited in the paper showed that when trees die (not directly related to an extreme event as embolism) they are largely depleted in NSC compared to living ones. In particular as NSC are not only used for developing leaves but also for a lot of defending processes. So there is a level of NSC under which trees cannot survive to any supplementary stress. So in

our representation a 0 value of R doesn't mean 'no NSC' but a value under which the probability of tree die becomes very important. Likewise it is true that carbon starvation is one of the causes for tree death but not the main one: xylem embolism for instance is also an important cause. Partial embolism is also a factor that can alleviate the capability of trees to grow after extreme events. This is the reason why we mention the text embolism, and not only of carbon starvation. What is called "reserve" in the model should then be viewed not only as carbohydrate reserves but more generally "what can impact the growth of trees". We thank the reviewer for this comment, as this was confusing in the text. This is better explained in the manuscript.

L50 - There needs to be a clear definition of 'ruin'. The farther I read the more convinced I am that this terminology needs to be more clearly set.

Thanks for the comment, as indeed the concept applied to forest needs to be clarified. Ruin in the context of the paper means that the carbon reserve is no longer sufficient for the growth, so that the majority of trees are dying. This concept of ruin can be applied to a forest in general but also to a given tree species, which means in this case that this species will disappear from the forest but that a forest with other more tolerant species can still exist. The definition is clarified in the introduction of the revised manuscript

L52 - What is meant by 'disappearance of trees'? There death and total respiration?

Thanks for the comment. The term was not clear: this just means tree death. It is corrected

L55 - what does 'average capital' mean here?

It means "average carbon reserve". This is corrected.

L59 - how is hazard defined in the context of this paper? It is a term that has many interpretations.

"Hazard" means: a potential source of harm (see e.g. https://en.wikipedia.org/wiki/Hazard or IPCC SREX 2012). In that instance (end of introduction), we actually meant "damage function from climate hazards".

L65 - Please spend some time making this more intelligible to readers from the natural sciences. Pretty much nobody who reads ESD will come from a financial background so it is worth the word count to better expand on the terminology. E.g.'balance between competing companies' - what companies? 'the capital vanishes' - whose?

OK. This will be clarified (see point L79 below).

Eqn 1 - shouldn't pt have the t subscripted?

No: it is p \times t. A fixed premium rate p is collected each year t. Of course, p could also depend on t (which is what we do in Eq. (6)). We will add a $\cdot$ between p and t in Eq. (1) to clarify this.

L71 - I suggest dropping the 'horizon' terminology. This might be a carryover from the insurance models but this is not a common way to talk about this in ecosystem modelling.
The summary for policymakers of the IPCC AR5 (1st chapter; https://www.ipcc.ch/srocc/chapter/summary-for-policymakers/) states:

"C.1.1 The temporal scales of climate change impacts in ocean and cryosphere and their societal consequences operate on time **horizons** which are longer than those of governance arrangements (e.g., planning cycles, public and corporate decision making cycles, and financial instruments)."

We use "horizon" with the same meaning of the IPCC AR5. Therefore, we think that it is appropriate to use "horizon" when speaking of a time bound in an ESD paper. We made this clear in the manuscript, so that the readers who are not familiar with IPCC language do not think that it is insurance jargon, which we do not speak.

L79 - other insurance companies bidding for the same clients to sell insurance to?- Small point, consider the tenses used. There are several instances of future tense that don't make sense. They make the reader wonder if this is going to appear in some future paper or ?

We meant to say that, as insurance companies compete with each other to get clients (i.e., you or us), they have to find a balance between a high premium rate (to ensure an income for them) and attracting clients from other companies (i.e. being somewhat cheaper). The necessity for this economic balance gives an upper bound for the premium rate. In other words, having a very high premium rate might not solve the problem of ruin for insurance companies, because they would lose all their clients to less greedy companies. This basic concept of capitalist economy will be clarified in the text, although it is marginally important for the paper.
The future tenses will be checked.

L102 - does this mean you don't let them allocate resources to their stems? So they can't grow?

No: obviously it means allocation to all the organs of the plants, so this is corrected.

L 104 - Fix the cite Handeregg.

OK. Sorry for the typo. This is corrected.

L108 - does the p0 change spatially?

Here we considered the model on a given region, and in the simplified application presented in the paper we assume a constant NPP, but obviously if this model is applied regionally p0 can change spatially (as well as the others model parameters) and could depend on the tree species. This is stated in the revised version.

 How is it determined?

p0 is heuristically chosen so that Rmax is reached in 4 years when no hazard occurs (p0=25). This choice is constrained by tree species. Following the two reviewers' remarks we will revise the different parameters (including p0) to be defined from literature and then better represent real cases. This is clarified in the manuscript.

 Does B have an upper limit? Assumedly it is constrained such that p(t) >= 0 as I am not sure what a negative p would mean since the loss is supposed to come from the S term.

Indeed: if trees do not shrink (negative p), an upper bound for B would be $p_0/E(S)$, where $E(S)$ is the mathematical expectation of $S$.

L118 - is there any ref you can use for proof of this here?

This is a classical result in statistics: the probability distribution of the number of times that a time series exceeds a high threshold converges to a Poisson distribution. This will be clarified in the text, with a reference to classical but pedestrian textbooks (e.g. Coles, S. An introduction to statistical modeling of extreme values. Springer series in statistics. London, New York: Springer, 2001.), and references in the atmospheric science literature (e.g. Smith, R. L., and T. S. Shively. Point process approach to modeling trends in tropospheric ozone based on exceedances of a high threshold. *Atmospheric Environment* 29(1995): 3489-99.), although google scholar provides hundreds of such uses in atmospheric sciences.

L155 - Can it be expanded upon how to estimate these parameters from observations? It is one thing to suggest the possibility but I think it is more helpful to try and relate these parameters to something more grounded.

Our sentence is indeed vague. $A_h$ and $B$ relate how climate hazards impact the growth of trees. As a first approximation, they could be obtained from a correlation between the climate conditions during identified droughts (e.g. 1976, 2003, 2018,

2019) and tree growth parameters (e.g. tree ring width/density, NPP), which are determined in situ of from satellite observations (NPP). We believe that obtaining such a correlation/regression is a whole methodological paper in itself. This important point will be discussed in the revised manuscript. Here, we take a rough approximation from He W, et al. (Patterns in nonstructural carbohydrate contents at the tree organ level in response to drought duration. Glob Chang Biol. 2020 Jun;26(6):3627-3638. doi: 10.1111/gcb.15078) for beech, which scales the maximum damage to ~20% of the maximum reserve.

Section 2.2 - this whole section is a bit abstract. It would be beneficial to have included a toy example. Even a simple financial one where we could see how these equations play out. I would like to see that included especially with a figure. I think that would benefit the paper and help the reader wrap their head around these concepts. After all,this is the first paper to intro the concept in the eco modelling lit.

Section 2.3 (sample trajectories) just does that by illustrating what trajectories look like.

L156 - So what units would all of these have? I am unsure what an Rmax means,100%? If %, then of what?

A percentage is indeed more meaningful, although this would not change the gist of the paper (see point below). This is explained in the revised manuscript.

L 164 - 'reserve units' - this seems like we should be able to use real units here.

Rmax represents the upper limit of tree "reserves" as tree do not tend to accumulate reserves indefinitely. If a sufficient level is reached, the plant allocates its remaining assimilate to growth of different organs. Concerning the units, if we considered that R represents stricto sensu carbohydrate reserve we should indeed use gC.m2 for instance. However as discussed previously and noticed by the reviewer, considering only carbohydrate reserve as driver of tree growth and tree die is too restrictive as others processes as xylem embolism should also be considered. So in such a theoretical model, "reserve" should be considered as all "that allow trees to initiate growth on next year" and then cannot be associated to a specific unit and then 100 is an arbitrary unit scaled with the others parameters of the model and calibrated to give reasonable results compared to observations. This is now explained in the text.

L 161 - is this missing an "and" so would be 'and one of the trajectories with a ruin'?

OK. An "and" is added.

Fig 1 a,b - please make it so that it is possible to get some info from this figure. Have multiple y-axis (sep plots) as right now it is not useful. Also I don't really understand

how this works. Since it is 5-median-95 then I understand why the median is above the 5 and 95. But this looks like the actual 95 quantile realization was chosen (rather than the representative behaviour). Why not choose the mean and then give us the average behaviours? Right now it just looks like so much noise. As far as I can tell this figure is trying to make an important point that the model can capture differences due to aniso/iso strategies, so I think it is worth the effort to make it more convincing.

Sorry: there was typo. The caption is corrected. The upper panels were indeed hard to read. We now chose to show trajectories of S(t) with a finite time ruin (in red) and the one that achieves the median R(t). The overall statistical properties of S(t) are (by construction) the same across all members of the simulations in Fig. 1. The new panels a and c illustrate that the "cluster" of hazards provokes the ruin.

L177 - The model used here doesn't equate any reserves to stem (line 102). Is there any paper to point to that has directly linked the two? Does the Cailleret paper then do that? As written that isn't clear.

In the text we mention only allocation to roots and leaves because these are the organs that play a direct role in growing. But obviously it does mean that reserves are not allocated to other parts of the plant as stems and trunk. But to avoid confusion, we corrected this in the text.

L181 - But aren't the simulations showing the decrease in reserves and not growth?If you are equating the reserves to growth, what does it mean when they trend to 0? No growth for a tree doesn't necessarily mean death but earlier that seems to be what is suggested (line 52). This whole paragraph is playing it very loose with terminology and relating poorly defined components of the ruin model with different real world observations. This needs to be tightened up considerably to be made consistent.

Thanks for the comment. Actually, a living tree cannot have null growth anyway. However, Indeed growth is obviously different from reserves. In particular with the same level of non carbohydrate reserves (NSC) the tree ring growth will depend on the climate condition of the year. But considering the relative difference between tree ring growth of healthy trees and dying trees is used as a proxy of NSC in papers like Caillerets et al. since it allows to disentangle the effect of climate and then it is well correlated to NSC status (as it is not possible to directly measure NSC of trees already dead). In the model, we also consider a relative change in reserve status (as mentioned before Rmax is fixed to an arbitrary value of 100), the comparison between the change in R and the relative change in tree ring growth make qualitatively sense. This will be detailed in the text.

Fig 2 could be made into a table and my suggested toy example be added as a figure.Fig 2 has little interesting information that a table couldn't show.

OK. This figure was meant to reassure statisticians, but Table 1 summarizes the same information. Fig. 2 was removed.

Table 1 - HW = heatwave?

Yes. This is clarified in the revised manuscript.

L239 - These numbers make me think you should then be able to go into the literature to find out how reasonable these are. Are there any reports that would substantiate what your model has found?

This type of estimate is coherent with other studies (e.g. Parey et al. Validation of a stochastic temperature generator focusing on extremes, and an example of use for climate change. *Climate Research* 59, n° 1 (2014): 61-75; Kharin et al. Changes in temperature and precipitation extremes in the CMIP5 ensemble. *Climatic change* 119, n° 2 (2013): 345-57).

L241 - repeated text that makes it confusing.

L241 did not repeat any text, as it presents a new result (with fixed hazard parameters).

Fig 4 - how is the avg reserve before ruin >0 when ruin was defined as R(t) = 0 (lines70, 101)? How much before ruin is used in the calculation? I think this needs a time period defined and specified. Is 4e meant to have ruin year for the y axis label?General - I would suggest that instead of 'cash' and 'credit' the terms be more ecological like 'aniso' and 'iso', it would help the reader place into context.

First: not all trajectories reach a ruin. And the average reserve (before ruin) is always positive (the average of any number of positive numbers is positive). The average before ruin is the average of R(t) when t<\tau, and \tau is the ruin time.
We rephrased the terminology to a more ecological context, following your suggestion.

L 244 - Since the stat significance is mentioned here. Would it make sense to indicate in the figure which differences were significant?

A new figure was made (with other physiological parameters). Differences are clearer now.

L248 - I would not use 'globally' but rather something like 'On the whole'. Globally can be read as in, well, globally.

OK.

L256 - It does provide an estimate for sure, but I see little attempt here to evaluate the estimates. Can more effort be putting into evaluating the differences between the two strategies and whether any observations support the model results?

OK. The beginning of the conclusion was reformulated.

Refs cited:
Piper, F. I.: Drought induces opposite changes in the concentration of non-structuralcarbohydrates of two evergreen Nothofagus species of differential drought resistance,Ann. For. Sci., 68(2), 415–424, 2011.
Rosas, T., Galiano, L., Ogaya, R., Peñuelas, J. and Martínez-Vilalta, J.: Dynamics of non-structural carbohydrates in three Mediterranean woody species following long-term experimental drought, Front. Plant Sci., 4, 400, 2013.
Rowland, L., da Costa, A. C. L., Galbraith, D. R., Oliveira, R. S., Binks, O. J., Oliveira, A.A. R., Pullen, A. M., Doughty, C. E., Metcalfe, D. B., Vasconcelos, S. S., Ferreira, L. V., Malhi, Y., Grace, J., Mencuccini, M. and Meir, P.: Death from drought in tropical forests is triggered by hydraulics not carbon starvation, Nature, doi:10.1038/nature15539,2015

**Anonymous Review #2**

We thank the referee for the efforts devoted to this review. The comments of the reviewer are in red, our replies are in blue.

The submitted manuscript describes the application of the Cramer-Lundberg ruin model which is well-established in the insurance sector to tree mortality caused by droughts on 5 sites in Europe. It aims at introducing this model to climate and Earth system science to enable straightforward support for decision-making, something – as the authors claim – the tipping-points lacks. Because it is a simple model of tree mortality, tested at 5 climate stations in Europe, which describes the climate hazard events,
I was wondering whether it would be more appropriate to transfer the manuscript to Natural Hazards and Earth System Science. In my view, the manuscript is lacking the feedback and resilience analysis and thus true interdisciplinary research to fit to the scope of ESD. Furthermore, the manuscript is not well developed that it sets its new idea of applying the Cramer-Lundberg model to quantify tree mortality into the context of existing literature on modelling tree mortality due to drought (the climate hazard) under current and future climate change. It is hastily written and not sufficiently substantiated by the body of literature which is essential when introducing a new concept.

We do have a different view of interdisciplinarity (in this paper: putting together ideas from econometrics and earth sciences). We feel that our manuscript seems more interdisciplinary than most recently published papers in ESD, including from the editors-in-chief of the journal.

I describe my major concerns in the following:
1) The introduction motivates the study with claims that a) the ecosystem service literature ignores the fact that ecosystem services are also threatened by disturbances or hazards,

We do not understand this point raised by the reviewer as we never claimed that ecosystem service literature ignores disturbances and hazards. Moreover, our paper is not about ecosystem services (although this term was mentioned). The introduction was rewritten.

and b) tipping points are mostly qualitative, not providing probabilities, and policy makers make little use of such studies. Several problems arise with these claims. a. For a) the claim is simply not true, the ecosystem service literature does recognize climate extremes, incl. fires and drought, as disservices (see e.g. (Shackleton et al., 2016)). Further, the authors claim that it is a dogma that ecosystems provide services to society. I am not sure if the term "dogma" is a polemic claim or a

misunderstanding from not translating it into a corresponding English term. The global IPBES assessment (Diaz et al., 2019) reflects the scientific agreement of an international body of scientists that this is the case. b. For b) Lenton et al. (2008) does provide the time scales at which the tipping points would occur and the literature on tipping points increasingly defines or refines those thresholds, e.g. Hirota et al. (2011) or Zemp et al. (2017) for the Amazon tipping element, or the Antarctic ice sheet (Garbe et al., 2020). Furthermore, it is not explained which limitations the tipping point concept has to answer the questions this paper aims to answer.

We rephrased the introduction. As stated in the title, we focus on trees, not ice sheets. We opted to cite seminal papers, which contain the main ideas, while the newer ones are essentially applications of existing tools and concepts. Other recent papers on forest tipping points are now cited.

2) The introduction of collapsology to the Earth System Science community is not thoroughly done. One 15-year-old citation is provided in the introduction which is not sufficient to introduce the ESD readership to this scientific field which is unknown to this community.
Again here, the state-of-the-art of this concept is not well described and the scientific gap not well developed. Furthermore, it is lacking a clear description of why a new concept is needed (things the ecosystem service concept cannot answer and the tipping point concept does not deliver), and why exactly this proposed concept is expected to provide a better solution.

We did not mean to write an introduction to collapsology, but merely related our study to that concept. We removed the discussion on collapsology in the introduction, which could sound far fetched.

3) The claims on the decision-making literature (from line 25) is not supported by literature, so lacks evidence. The authors need to provide evidence or overview on how the tools established in insurance and finance provide "all the tools for decision-making", examples must be provided here to substantiate this claim.

The reviewer seems to overinterpret our claim. We obviously do not claim that the insurance sector provides *all* tools for *all* decisions: we claim that the insurance sector has developed all tools that are necessary to forecast its own losses and benefits. Such tools seem to be  accepted (at least implicitly) by a majority of humans who are willing to pay a fee to insure their health, house or car. We believe that this fact speaks for itself. The sentence was reformulated to clarify this.

4) The paper then later on does not get back to a tipping point/resilience or close collapse analysis nor does it make use of the ecosystems service-disservice

concept. The authors do not get back to the issues raised in the introduction. This also applies to the decision-making tools mentioned in the introduction.

Indeed. This is why the introduction was rewritten. Thank you for those comments. The paper is not about ecosystem service or disservice. Such terms were removed from the text.

5) The general assumption is that the ruin of ecosystems can be captured with tree mortality. And tree mortality does not capture all patterns and processes of an ecosystem. This is an oversimplification that affects the outcome and interpretation of results of the study. Well, it only applies to wooded ecosystems. In addition to the description of forests affected by drought must be accompanied by an explanation on how the collapsology concept can be transformed to Earth system science, specifically ecosystem dynamics. This is the missing link which needs to be explained to correctly set the scene. An ecosystem is more complex than paying something in (GPP) and losing something (due to drought). So, the paper does not provide the evidence why the Cramer-Lundberg model or its extension is a better description of processes leading to drought-related tree mortality.

We never claim that the ruin of ecosystems (in general) stems from tree mortality. We focus on tree mortality which arises when they lose the ability to grow (which we call "ruin") . Then we claim that this concept of ruin modelling can be transposed to other ecosystems, which would obviously require a specific model adjustment.

For example, the paper does not address the question of tipping points in herbaceous ecosystems which have a very different behavior in response to climate hazards. We make this more explicit in the revised manuscript.

6) If the model has to produce 104 sample members, and it is shown for 5 meteorological stations only, I doubt its computational costs if applied to the global scale for a range of climate scenarios.

Indeed, reconstructing a probability distribution requires producing a very large number of simulations, which would not be possible with a complex process-based model, but which is not a problem with the ruin model as obtaining the Generalized Pareto Distribution parameters of any variable at a global scale (i.e. for all grid points of a climate model) is a matter of minutes on a PC, and a few seconds on a parallel computer (e.g., Kharin, V.V et al. Changes in temperature and precipitation extremes in the CMIP5 ensemble. Climatic change 119, no 2 (2013): 345-57), and is done only once. Doing our 10000 Monte-Carlo simulations from those parameters for Europe takes a couple of minutes on a PC. Doing it at a global scale and with high resolution would take a few hours on a PC (at most), a few minutes on a supercomputer, as computations can be (are) parallelized.

7) It is not explained why a new drought index had to be developed and why not existing and well-established drought indices could be used. This is important and missing in the manuscript.

A new drought index is indeed marginally important for the ruin model itself, which is why it was deferred to an appendix. What is important in the ruin model are the parameters of the probability distribution of hazards (intensity, duration and frequency). Well-established drought indices (such as the one of de Martone) are not physically satisfactory, as explained in the appendix. There are other more physically satisfactory drought indices, which do account for physical processes, but they cannot be estimated over long periods of time, and do not allow a reliable estimate of parameters for the hazard model. This motivated this new drought index, which can be computed with basic climatic variables that are recorded over long periods of time. This is clarified in the revised manuscript.

8) Drought occurrence is not a random process. The assumption for S(t) needs to be revised. Plants have more adaptation mechanisms by which they can avoid carbon starvation, loss of productivity (GPP) due to closed stomata and increased maintenance respiration. They have evolved physiological strategies and physiognomic structures to avoid transpiration loss. It can't be subsumed with having a carbon reserve pool or not. I can understand why this cannot be implemented in a simple model, but some notification of this knowledge is required to justify the model assumptions.

We do not agree with the preamble of this comment. The macroscopic phenomenological development of a drought (or any climate hazard) is indeed a deterministic process, but key quantities like frequency, duration or intensity can be modeled by random processes. This is the core of statistical climatology.

9) Lines 92-94: unclear how this can be transformed to the tree-mortality application. This needs to be described here. Also, how this can help to advance science wrt drought impacts on increasing tree mortality and the stability of ecosystems.

Our claim (in those lines 92-94) is that it is possible to determine the probability distribution of ruin time from statistical properties of hazards. The following section (2.2) shows how the Cramer-Lundberg model (and the surrounding probabilistic framework) can be transposed to tree-mortality (that we generically call "ruin"). This is more explicit in the revised text.

10) Line 105, NPP needs to be properly introduced. Totally open, and not explained, how $p_0$ for the investment of NPP to the reserve pool can be justified.

OK. The acronym NPP (=net primary production) is now provided at its first occurrence. The choice of the value of p_0 is now explained.

11) Line 104: what is the damage function? S(t) was introduced with a different meaning.

S(t) is a "damage function" due to climate hazards. This terminology was streamlined between sections 2.1 and 2.2.

12) It needs to be shown that the climate data, i.e. number of droughts, indeed are Poisson and GPD distributions.

Under rather generic mathematical conditions, the probability distribution of the exceedances over a threshold can be modeled by a Generalized Pareto Distribution. This is analogous to the fact that the average of a variable that has a finite standard deviation can be modeled by a Gaussian. The principle is the same for the frequency of exceedances (or inter-arrival times). The simplest statistical model that describes the time interval between exceedances of a threshold is a Poisson distribution. We have recalled this in the revised manuscript by citing textbooks that are often used in statistical climatology or hazard models (e.g. S. Coles, An Introduction to Statistical Modeling of Extreme Values, Springer, 2001).

13) Line 155: the authors need to provide evidence that the parameter from their model can indeed be directly measured and evaluated using observations. This statement is not substantiated by evidence.

The parameters were initially chosen arbitrarily, in order to illustrate the contrasting behavior of "cash" and "credit" strategies. This has been mentioned in the discussion of the paper. In the revised version, we re-did analyses that use constraints obtained from the literature, e.g. He W, et al. (Patterns in nonstructural carbohydrate contents at the tree organ level in response to drought duration. Glob Chang Biol. 2020 Jun;26(6):3627-3638. doi: 10.1111/gcb.15078), who performed a meta-analysis of reserve reduction in case of severe drought.

14) The findings that trees die at the time scale of decades to 100 years, is widely known and evidence is provided. The question is rather, if the model can produce increased drought-related mortality 3-5 years after a severe drought and the authors need to show how their findings compare to other model results or estimates based on drought-indices. There is an ample body of literature that has to be referenced here. Specifically, the result in line 182 indicates age mortality and not something related to a drought hazard.

Thank you for this comment. Indeed, very few trees in Europe live longer than a century. We consider that our study applies to a collection of trees (e.g. a forest), whose lifetime is hoped to be longer than a century. This is clarified in the revised text.

15) Validation of modelled results is not provided and needs to be included.
Although we appreciate this comment (also made by referee#1), we would like to point out that since the seminal paper of E.N. Lorenz (Deterministic nonperiodic flow. J. Atmos. Sci. 20 (1963): 130-41), many studies, including publications in ESD, have focused on the behavior of idealized models.

References
a. Diaz, S., Settele, J., Brondizio, E. S., Ngo, H. T., Agard, J., Arneth, A., Balvanera, P., Brauman, K. A., Butchart, S. H. M., Chan, K. M. A., Garibaldi, L. A., Ichii, K., Liu, J., Subramanian, S. M., Midgley, G. F., Miloslavich, P., Molnar, Z., Obura, D., Pfaff, A., Polasky, S., Purvis, A., Razzaque, J., Reyers, B., Chowdhury, R. R., Shin, Y. J., Visseren-Hamakers, I., Willis, K. J., and Zayas, C. N.: Pervasive human-driven decline of life on Earth points to the need for transformative change, Science, 366, 10.1126/science.aax3100, 2019.
b. Garbe, J., Albrecht, T., Levermann, A., Donges, J. F., and Winkelmann, R.: The hysteresis of the Antarctic Ice Sheet, Nature, 585, 538-544, 10.1038/s41586-020-2727-5, 2020.
c. Hirota, M., Holmgren, M., Van Nes, E. H., and Scheffer, M.: Global resilience of tropical forest and savanna to critical transitions, Science, 334, 232-235, 10.1126/science.1210657, 2011.
d. Shackleton, C. M., Ruwanza, S., Sinasson Sanni, G. K., Bennett, S., De Lacy, P., Modipa, R., Mtati, N., Sachikonye, M., and Thondhlana, G.: Unpacking Pandora's Box: Understanding and Categorising Ecosystem Disservices for Environmental Management and Human Wellbeing, Ecosystems, 19, 587-600, 10.1007/s10021-015-9952-z, 2016.
e. Zemp, D. C., Schleussner, C. F., Barbosa, H. M. J., and Rammig, A.: Deforestation effects on Amazon forest resilience, Geophysical Research Letters, 44, 6182-6190, 10.1002/2017gl072955, 2017.

We thank the referee for those interesting recent references.

---

## Editor Decision (ED1)

**Modelling the Ruin of Forests under Climate Hazards**

Pascal Yiou[1] and Nicolas Viovy[1]

[1]Laboratoire des Sciences du Climat et de l'Environnement, UMR 8212 CEA-CNRS-UVSQ, IPSL & U Paris-Saclay, 91191 Gif-sur-Yvette Cedex, France

**Correspondence:** P. Yiou (pascal.yiou@lsce.ipsl.fr)

**Abstract.** Estimating the risk of collapse of forests due to extreme climate events is one of the challenges of adaptation to climate change. We adapt a concept from ruin theory, which is widespread in econometrics or the insurance industry, to design a growth/ruin model for trees, under climate hazards that can jeopardize their growth. This model is an elaboration of a classical Cramer-Lundberg ruin model that is used in the insurance industry. The model accounts for the interactions between physiological parameters of trees and the occurrence of climate hazards. The physiological parameters describe interannual growth rates and how trees react to hazards. The hazard parameters describe the probability distributions of occurrence and intensity of climate events. We focus on a drought/heatwave hazard. The goal of the paper is to determine the dependence of ruin and average growth probability distributions as a function of physiological and hazard parameters. From extensive Monte Carlo experiments, we show the existence of a threshold on the frequency of hazards beyond which forest ruin becomes certain in a centennial horizon. We also detect a small effect of strategies to cope with hazards. This paper is a proof-of-concept to quantify collapse (of forests) under climate change.

**1 Introduction**

 Extreme events such as droughts and heatwaves are climate hazards that have short and long term effects on forests. The recent accumulation of drought/heat stress to forests might lower their resilience to future extreme events (e.g. Wigneron et al., 2020; Flach et al., 2018; Bastos et a . It has been observed that such events increase the chance of tree mortality (Anderegg et al., 2013). Such an effect questions the survival of tree species in some regions of the world (Zeppel et al., 2011; Lindenmayer et al., 2016). The mechanisms leading to tree mortality include complex physiological processes that can depend on the tree species and type of hazards (e.g. Choat et al., 2012).

Most studies on tree death are based on direct or indirect observations of the tree behavior and their growth parameters. They give precious information on the observed global response of forests to those climate hazards. But they are by essence

limited to the length of the observation period, and have mostly focused on a few key observed events (e.g. Rao et al., 2019) which can hinder a statistical description that would be necessary to build projections, or estimate *risk* (Field et al., 2012) as response to climate change. Here *risk* is understood as the probability distribution of a failure (e.g. irreversible damage or death) due to climate hazards (Katz, 2016).

The potential disappearance of whole forest or a given tree species, due to changes in climate features can be qualified as a *tipping point* of climate change. There has ample literature on tipping points of the climate system, i.e. climate thresholds beyond which ecosystems change behavior (Lenton et al., 2008; Levermann et al., 2012). Those papers have defined methodologies to identify climate thresholds beyond which ecosystems are endangered. They have been used to infer tipping points of forests (Reyer et al., 2015; Pereira and Viola, 2018).

The key concept of this paper is the use of the so-called *ruin theory* to provide quantitative elements on the probability of such tipping points. There many papers in the econometrics literature that describe ruin models for insurance and finance since the seminal work of Lundberg (1903). A mathematical and statistical (Asmussen and Albrecher, 2010) has helped determining the optimal parameters of such models, so that insurers or investors limit the risk of losing their investment and maximize their gain. Ruin models are used to describe the probability that a company that grows with regular income can get bankrupt due to external hazards. To the best of our knowledge, this literature has never been to environmental sciences.

The rationale of the choice of trees or forests in this paper is that, as opposed to annual plants, trees are adapted to live for a long time and then should be able to survive to bad climate conditions on a given year. Mechanistic growth models for trees can be devised (e.g. Han and Singh, 2020) and observations of tree mortality are available (e.g. Choat et al., 2012). Tree growth is affected by climate variations in various ways. Heatwaves and droughts can alter (and lower) tree reserves and the capability for growth during the years following event, which affects their average growth and can increase their chance of mortality (von Buttlar et al., 2017; Sippel et al., 2018).

60 ~~A parallel between insurance system and trees can be done. As opposed to annual plants, trees are adapted to live for a long time and then should be able to avoid death related to damage due to bad climate conditions during a year. Those damages can be related to a reduced productivity and in the worst cases to a destruction of a part of the trees (related for instance to xylem embolism that kills the branches).theycarbohydrate and a structure (branches)thatbuildlarge surface of~~ leaves (po-

[revised manuscript text omitted]

---

## Author Response (AR2)

Reply to Editor's comments (our reply is in blue)

**Dear authors,**
Dear Editor,

**Thank you for revising your manuscript following the reviewers' comments. Given the nature of comments from both reviewers, I took the time to very carefully go through the entire manuscript myself.**

**Overall your manuscript now reads much better than before. I have suggested mostly minor, but multiple, changes in the attached annotated PDF file of your manuscript. These are mostly English language related changes. English is not my first language so please use your judgement as well. I think these changes will overall make the manuscript more readable and easy to follow.**

**There are few places where I have also requested more clarification. I look forward to reading your revised manuscript.**

**It was difficult for me to suggest changes on a PDF file. Please use the free Adobe Acrobat reader to open the file and then open the "Comments". This will open a panel on the side which will make it easier to see all my comments.**

**Thanks,**
**Vivek.**

Thank you for having taken the time for this thorough verification. We went through the 197 entries of the annotated pdf file and provided point-by-point replies. When a typo is pointed out, we flagged the comment with "OK". A more detailed reply was given when appropriate. A track change file is appended to this page.

Best regards,
Pascal Yiou and Nicolas Viovy.

**Modelling the Ruin of Forests under Climate Hazards**

Pascal Yiou[1] and Nicolas Viovy[1]

[1]Laboratoire des Sciences du Climat et de l'Environnement, UMR 8212 CEA-CNRS-UVSQ, IPSL & U Paris-Saclay, 91191 Gif-sur-Yvette Cedex, France

**Correspondence:** P. Yiou (pascal.yiou@lsce.ipsl.fr)

**Abstract.** Estimating the risk of collapse of forests due to extreme climate events is one of the challenges of adaptation to climate change. We adapt a concept from ruin theory, which is widespread in econometrics or the insurance industry, to design a growth/ruin model for trees, under climate hazards that can jeopardize their growth. This model is an elaboration of a classical Cramer-Lundberg ruin model that is used in the insurance industry. The model accounts for the interactions between physiological parameters of trees and the occurrence of climate hazards. The physiological parameters describe interannual growth rates and how trees react to hazards. The hazard parameters describe the probability distributions of occurrence and intensity of climate events. We focus on a drought/heatwave hazard. The goal of the paper is to determine the dependence of ruin and average growth probability distributions as a function of physiological and hazard parameters. From extensive Monte Carlo experiments, we show the existence of a threshold on the frequency of hazards beyond which forest ruin becomes certain in a centennial horizon. We also detect a small effect of strategies to cope with hazards. This paper is a proof-of-concept to quantify collapse (of forests) under climate change.

*Copyright statement.* TEXT

**1 Introduction**

 Extreme events such as droughts and heatwaves are climate hazards that have short and long term effects on forests. The recent accumulation of drought/heat stress to forests might lower their resilience to future extreme events (e.g. Wigneron et al., 2020; Flach et al., 2018; Bastos et a . It has been observed that such events increase the chance of tree mortality (Anderegg et al., 2013). Such an effect questions the survival of tree species in some regions of the world (Zeppel et al., 2011; Lindenmayer et al., 2016). The mechanisms leading to tree mortality include complex physiological processes that can depend on the tree species and type of hazards (e.g. Choat et al., 2012).

Most studies on tree death are based on direct or indirect observations of the tree behavior and their growth parameters. They give precious information on the observed global response of forests to those climate hazards. But they are by essence

 limited to the length of the observation period, and have mostly focused on a few key observed events (e.g. Rao et al., 2019) which can hinder a statistical description that would be necessary to build projections, or estimate *risk* (Field et al., 2012) as response to climate change. Here *risk* is understood as the probability distribution of a failure (e.g. irreversible damage or death) due to climate hazards (Katz, 2016).

The potential disappearance of whole forest or a given tree species, due to changes in climate features can be qualified as a *tipping point* of climate change. There has been ample literature on  tipping points of the climate system, i.e. climate thresholds beyond which ecosystems change behavior  (Lenton et al., 2008; Levermann et al., 2012, just to cite seminal papers).  papers have defined methodologies to identify climate thresholds beyond which ecosystems are endangered. They have been used to infer tipping points of forests (Reyer et al., 2015; Pereira and Viola, 2018).

~~A body of literature on so-called *collapsology* has emerged in the past few years (Diamond, 2005). Those studies describe mechanisms that would make the key institutions of society collapse due to an accumulation of big or small events. Again, those considerations are generally qualitative as to "when" and "how intense", as they address very general issues and no mechanistic model of a collapsing system is analyzed.~~

 The key concept of this paper is the use of the so-called *ruin theory* to provide quantitative elements on the probability of such tipping points. There  many papers in the econometrics literature that describe ruin models for insurance and finance since the seminal work of Lundberg (1903). A mathematical and statistical  (Asmussen and Albrecher, 2010) has helped determining the optimal parameters of such models, so that insurers or investors limit the risk of losing their investment and maximize their gain. Ruin models are used to describe the probability that a company that grows with regular income can get bankrupt due to external hazards. To the best of our knowledge, this literature has never been  to environmental sciences.

 The rationale of the choice of trees or forests in this paper is  that, as opposed to annual plants, trees are adapted to live for a long time and  should be able to survive to bad climate conditions on a given year. Mechanistic growth models for trees can be devised (e.g. Han and Singh, 2020) and observations of tree mortality are available (e.g. Choat et al., 2012). Tree growth is affected by climate variations in various ways. Heatwaves and droughts can alter (and lower) tree reserves and the capability for growth  during the years following  event, which affects their average growth and can increase their chance of mortality (von Buttlar et al., 2017; Sippel et al., 2018

60 ~~A parallel between insurance system and trees can be done. As opposed to annual plants, trees are adapted to live for a long time and then should be able to avoid death related to damage due to bad climate conditions during a year. Those damages can be related to a reduced productivity and in the worst cases to a destruction of a part of the trees (related for instance to xylem embolism that kills the branches).theycarbohydrate and a structure (branches)builda large surface of~~ leaves (po-

65 tential productivity  is related to this foliage area)   the next year; they  
[revised manuscript text omitted]

---

## Author Response (AR3)

Reply to the Editor's comments (our replies are in red).

Dear authors,
Dear Editor,

Thank you for making changes to your manuscript following my comments. As you know, I have been communicating with you outside the ESD system to accelerate resolving issues related to several items in your manuscript. Thank you for your help in resolving these issues. In the attached PDF file you will find these and other minor issues which need your attention.

Thank you. We attached the replies to the queries/comments in the pdf file and a manuscript with track change.

Please pay close attention to the figures and the issue related to units of p0 and R. I have also suggested to change "ruin" to "collapse" since ruin is a more insurance related word.

We strongly prefer to stick to "ruin" rather than "collapse":
- The second referee argued against references to "collapsology".
- The term "ruin" is necessary to introduce the Cramer-Lundberg model (on which our model is based), and it refers to "ruin theory", which is a well-established branch of statistical modelling. Our paper also argues that environmental sciences can benefit from advances in other sciences (like financial modeling), so that we believe that it is more efficient to use a terminology that already exists and can be related to a precise scientific domain.

Please feel free to contact me outside the ESD system if you have any questions to keep this process moving as fast as we can.

I look forward to reviewing the revised version.

Thanks,
Vivek.

Sincerely,
Pascal Yiou and Nicolas Viovy.

[revised manuscript text omitted]

---

## Author Response (AR4)

Reply to the Editor's comments (our replies are in red).

Dear Editor,

Thank you for your patience and making changes to your manuscript. As we discussed via email there are still some small issues that need your attention. These are summarized in the attached annotated PDF file.

We addressed the issues raised in the annotated pdf file, and attach a manuscript with track change.

Also, while one of the reviewers suggested the use of the word "indice", I think such a word doesn't exist in English. As far as I know, plural of index is indices (or indexes). But the word "indice" is incorrect. You may change it now or let the copy-editing folks catch it.

OK. "indice" will be changed to "index", while we keep the plural "indices".

Sincerely,
Pascal Yiou and Nicolas Viovy.

[revised manuscript text omitted]

---

## Author Response (AR5)

Reply to the Editor's comments (our replies are in red).

Dear authors,
Dear Editor,

Thank you for revising your manuscript following my feedback.

It seems you forgot to address two minor points.
We are sorry for this lack of attention, due to our work on laptop computers with small screens.
Both corrections have been made.

As far as I know, one normalizes a quantity BY something. So if I divide a quantity by it's mean

x(i)/mean(x)

that is written as

x(i) are normalized BY their mean.

Therefore, in my last review I requested that on line 165 in the latest version of your manuscript you replace "normalized to Rmax" by "normalized by Rmax".

Similarly, on line 395 please replace "normalizes cumulated precipitation and average temperature" by "normalizes cumulative precipitation by average temperature" since in the following

I_DM=P/(T+10)

P is divided by (T+10).

Finally, I think, "cumulated" should be "cumulative" but I will leave that for the copy-editors.

Again, we thank you for your vigilance.

I have suggested these as technical corrections so I don't need to see the manuscript again.

Sincerely,
Pascal Yiou and Nicolas Viovy.